# Fair Anomaly Detection For Imbalanced Groups

## Abstract

Anomaly detection (AD) has been widely studied for decades in many real-world applications, including fraud detection in finance, intrusion detection for cybersecurity, etc. Due to the imbalanced nature between protected and unprotected groups and the imbalanced distributions of normal examples and anomalies, the learning objectives of most existing anomaly detection methods tend to solely concentrate on the dominating unprotected group. Thus, it has been recognized by many researchers about the significance of ensuring model fairness in anomaly detection. However, in the imbalanced scenario where the unprotected group is more abundant than the protected group, the existing fair anomaly detection methods tend to erroneously label most normal examples from the protected group as anomalies. This phenomenon is caused by the improper design of learning objectives, which statistically focus on learning the frequent patterns (i.e., the unprotected group) while overlooking the under-represented patterns (i.e., the protected group). To address these issues, we propose FADIG, a fairness-aware anomaly detection method targeting the imbalanced scenario. It consists of a fairness-aware contrastive learning module and a rebalancing autoencoder module to ensure fairness and handle the imbalanced data issue, respectively. Moreover, we provide the theoretical analysis that shows our proposed contrastive learning regularization guarantees group fairness. Empirical studies demonstrate the effectiveness and efficiency of FADIG across multiple real-world datasets.

## 1 Introduction

Anomaly detection (AD), a.k.a. outlier detection, is referred to as the process of detecting data instances that significantly deviate from the majority of data instances (Chandola et al., 2009). Anomaly detection finds extensive use in a wide variety of applications including financial fraud detection (West & Bhattacharya, 2016; Huang et al., 2018), pathology analysis in the medical domain (Faust et al., 2018; Shvetsova et al., 2021) and intrusion detection for cybersecurity (Liao et al., 2013; Ahmad et al., 2021). For example, an anomalous traffic pattern in a computer network suggests that a hacked computer is sending out sensitive data to an unauthorized destination Ahmed et al. (2016); anomalies in credit card transaction data could indicate credit card or identity theft (Rezapour, 2019).

Up until now, a large number of deep anomaly detection methods have been introduced, demonstrating significantly better performance than shallow anomaly detection in addressing complicated detection problems in a variety of real-world applications such as computer vision tasks. For instance, Sohn et al. (2021); Li et al. (2023) aim to learn a scalar anomaly scoring function in an end-to-end fashion, while Audibert et al. (2020); Chen et al. (2021); Hou et al. (2021); Yan et al. (2021); Wang et al. (2023) propose to learn the patterns for the normal examples via a feature extractor.

Recently, there has been widespread recognition within the AI community about the significance of ensuring model fairness and thus it is highly desirable to establish specific parity or preference constraints in the context of anomaly detection. Take racial bias in anomaly detection as an example. Racial bias has been observed in predictive risk modeling systems to predict the likelihood of future adverse outcomes in child welfare (Chouldechova et al., 2018). Communities in poverty or specific racial or ethnic groups may face disadvantages due to the reliance on government administrative data. The data collected from these communities, often stemming from their economic

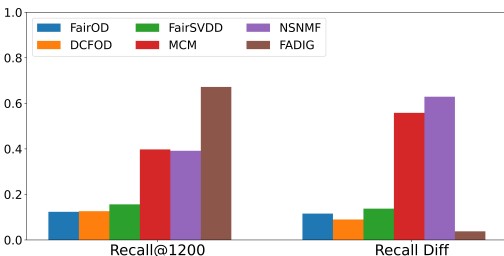

Figure 1: Recall@1200 and absolute Recall difference of the existing methods on the MNIST-USPS dataset.

| Methods | Unprotected | Protected |
|---|---|---|
| FairOD | 117(984) | 35(216) |
| DCFOD | 124(970) | 25(230) |
| FairSVDD | 141(741) | 44(459) |
| MCM | 238(327) | 234(873) |
| NSNMF | 196(294) | 267(906) |
| FADIG(ours) | 630(809) | 247(391) |

Table 1: True anomalies out of identified anomalies (number in the parentheses) of existing methods in each group on the MNIST-USPS dataset.

status and welfare dependence, can inadvertently categorize them as high-risk anomalies, leading to more frequent investigations to these minority groups. Consequently, disproportionately flagging minority groups as anomalies not only perpetuates biases but also results in an inefficient allocation of government resources.

To mitigate potential bias in anomaly detection tasks, numerous researchers (Song et al., 2021; Zhang & Davidson, 2021; Fioresi et al., 2023) advocate for incorporating fairness constraints into their proposed methods. However, in an imbalanced data scenario where the unprotected group is more abundant than the protected group, most of these methods tend to erroneously label most normal examples from the protected/minority group as anomalies. To better illustrate this issue, we provide a toy example on the MNIST-USPS dataset (Zhang & Davidson, 2021) where the size of the unprotected group is four times that of the protected group, and approximately 10% of the total samples are anomalies. Figure 1 and Table 1 show the performance of anomaly detection methods evaluated on this dataset, where *Recall Diff* refers to the absolute value of recall difference between the protected group and the unprotected group. Note that in such an imbalanced scenario, metrics such as accuracy difference (Zafar et al., 2017) are not proper choices. We observe that existing methods either compromise performance for fairness (i.e., low recall rate and low recall difference) or exhibit unfair behavior (i.e., high recall difference). The problem of misclassification arises from models focusing on learning frequent patterns in the more abundant unprotected group, potentially overlooking under-represented patterns in the protected group. The issue of group imbalance results in higher errors for protected groups, thus causing misclassifications. Following Hashimoto et al. (2018), we refer to this phenomenon as *representation disparity*.

To address these issues, we face the following two major challenges. **C1: Handling imbalanced data.** Due to the imbalanced nature between the protected and unprotected groups and the imbalanced distributions of normal examples and anomalies, the learning objectives of most existing anomaly detection methods tend to solely concentrate on the unprotected group. **C2: Mitigating the representation disparity.** Traditional anomaly detection methods encounter difficulties in dealing with representation disparity issues, which may worsen in the imbalanced data scenario as protected groups are typically fewer than unprotected groups.

To tackle these challenges, in this paper, we propose FADIG, a fairness-aware contrastive learning-based anomaly detection method for the imbalanced group scenario. FADIG mainly consists of two modules: (1) the fairness-aware contrastive learning module; (2) the re-balancing autoencoder module. Specifically, the fairness-aware contrastive learning module aims to maximize the similarity between the protected and unprotected groups to ensure fairness and address **C2**. In addition, we encourage the uniformity of representations for examples within each group, as ensuring uniformity in contrastive learning can be beneficial for the imbalanced group scenario (Jiang et al., 2021). To further address the negative impact of imbalanced data (i.e., **C1**), we propose the re-balancing autoencoder module utilizing the learnable weight to reweigh the importance of both the protected and unprotected groups. Combining the two modules, we design a simple yet efficient method FADIG with a theoretical guarantee of fairness. Our contributions are summarized below.

- A fairness-aware anomaly detection method FADIG addressing the representation disparity and imbalanced data issues in the anomaly detection task.

- Theoretical analysis showing that our proposed fair contrastive regularization term guarantees group fairness.

- The re-balancing autoencoder equipped with the learnable weight alleviating the negative impact of the imbalanced groups.

- Empirical studies demonstrating the effectiveness and efficiency of FADIG across multiple real-world datasets.

The rest of this paper is organized as follows. We first provide the preliminaries in Section 2 and then introduce our proposed fair anomaly detection method in Section 3, followed by the theoretical fairness analysis in Section 4. Then, we systematically evaluate the effectiveness and efficiency of FADIG in Section 5. We finally conclude the paper in Section 6.

## 2 PRELIMINARIES

In this paper, we explore the fairness issue in the unsupervised anomaly detection task. Among the various fairness definitions proposed, there is no consensus about the best one to use. In this work, we focus on the group fairness notion which usually pursues the equity of certain metrics among the groups. Without loss of generality, we consider the groups here to be the protected group and the unprotected group (e.g., Black and Non-Black in race). We are given a dataset $D = P \cup U$, where $P = \{x_i^P, y_i^P\}_{i=1}^n$ are examples from the protected group, $U = \{x_i^U, y_i^U\}_{i=1}^m$ from the unprotected group, and $x_i^P, x_i^U$ are sampled i.i.d from distributions $\mathcal{P}_P, \mathcal{P}_U$ over the input space $\mathbb{R}^d$ respectively. The ground-truth labels $y_i^P, y_i^U \in \mathcal{Y} = \{0, 1\}$ represent whether the example is an anomaly ($y = 1$) or not, which are given by deterministic labeling functions $a_P, a_U : \mathbb{R}^d \to \mathcal{Y}$, respectively. Note that we do not have access to the labels during training as we focus on the unsupervised anomaly detection setting.

The task of unsupervised anomaly detection is to find a hypothesis $h : \mathbb{R}^d \to \mathcal{Y}$ which identifies a maximal subset $\mathcal{A} \subset D$ whose elements deviate significantly from the normal examples in $D$. This identification is done without the aid of labeled examples, meaning the algorithm must rely on the intrinsic properties of the data, such as distribution, density, or distance metrics, to discern between normal examples and anomalies. The risk of a hypothesis $h$ w.r.t. the true labeling function $a$ under distribution $\mathcal{D}$ using a loss function $\ell : \mathcal{Y} \times \mathcal{Y} \to \mathbb{R}_+$ is defined as: $R_{\mathcal{D}}^\ell(h, a) := \mathbb{E}_{x \sim \mathcal{D}}[\ell(h(x), a(x))]$. We assume that $\ell$ satisfies the triangle inequality. For notation simplicity, we denote $R_P^\ell(h) := R_{\mathcal{P}_P}^\ell(h, a_P)$ and $R_U^\ell(h) := R_{\mathcal{P}_U}^\ell(h, a_U)$. The empirical risks over the protected group $P$ and the unprotected group $U$ are denoted by $\hat{R}_P^\ell$ and $\hat{R}_U^\ell$.

One direction of unsupervised AD is reconstruction-based autoencoder, such as An & Cho (2015); Audibert et al. (2020); Hou et al. (2021). Assuming the anomalies possess different features than the normal examples, given an autoencoder over the normal examples, it will be hard to compress and reconstruct the anomalies. The anomaly score can then be defined as the reconstruction loss for each test example. Formally, the autoencoder consists of two main components: an encoder $g_e : \mathbb{R}^d \to \mathbb{R}^r$ and a decoder $g_d : \mathbb{R}^r \to \mathbb{R}^d$, where $r$ is the dimensionality of the hidden representations. $g_e(x)$ encodes the input $x$ to a hidden representation $z$ that preserves the important aspects of the input. Then, $g_d(z)$ aims to recover $x' \approx x$, a reconstruction of the input from the hidden representation $z$. Overall, the autoencoder can be written as $G = g_d \circ g_e$, i.e. $G(x) = g_d(g_e(x))$. For a given autoencoder-based framework, the anomaly score for $x$ is computed using the reconstruction error as:

$$s(x) = \|x - G(x)\|^2, \tag{1}$$

where all norms are $\ell_2$ unless otherwise specified. Anomalies tend to exhibit large reconstruction errors because they do not conform to the patterns in the data as coded by the autoencoder. This scoring function is generic in that it applies to many reconstruction-based AD models, which have different parameterizations of the reconstruction function $G$. Next, we will present our method design based on the autoencoder framework. For quick reference, we summarize the notation used in the paper in Table 7 in Appendix.

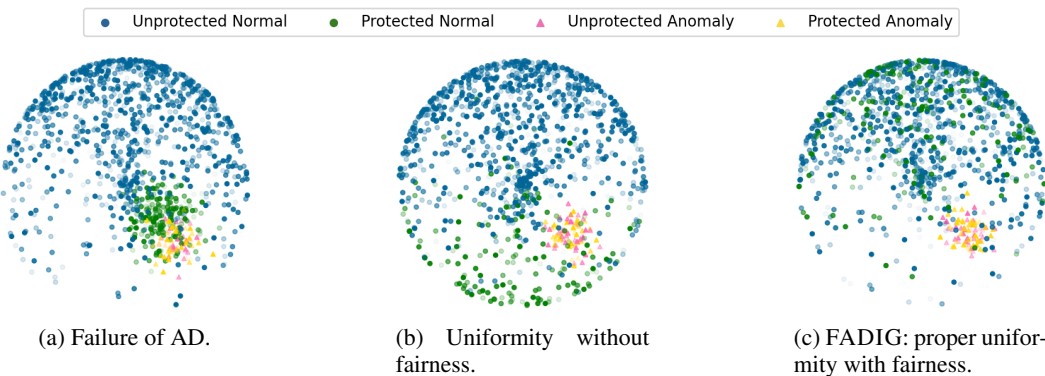

    (a) Failure of AD.       (b)  Uniformity  without       (c) FADIG: proper unifor-
                                                 fairness.                   mity with fairness.

Figure 2: Illustrations of uniformity. The blue and green circles denote normal examples from the unprotected group and protected group respectively. The pink and yellow triangles denote anomalies from the unprotected group and protected group respectively. The three subfigures illustrate three different projections from the same data set. With projection (a), many existing AD methods overly flag the examples from the protected groups (green circles) as anomalies (triangles). In projection (b), traditional contrastive regularization methods encourage uniformity but do not consider group fairness. In (c), our FADIG ensures group fairness while maintaining proper uniformity.

## 3   PROPOSED METHOD

Our proposed FADIG mainly consists of two modules: a Fairness-aware Contrastive Learning Module and a Re-balancing Autoencoder Module.

### 3.1   FAIRNESS-AWARE CONTRASTIVE LEARNING

Existing anomaly detection models (Song et al., 2021; Zhang & Davidson, 2021; Fioresi et al., 2023) statistically focus on learning the frequent patterns (i.e., the unprotected group), while overlooking the under-represented patterns (i.e., the protected group) within the observed imbalanced data. Due to the lower contribution of protected groups to the overall learning objective (e.g., minimizing expected reconstruction loss), examples from the protected groups may experience systematically higher errors. Thus, they tend to erroneously label most normal examples from the protected group as anomalies, producing unfair outcomes as shown in Figure 2a.

Recent works (Wang & Isola, 2020; Sohn et al., 2021) have shown that encouraging uniformity with contrastive learning can alleviate this issue by pushing examples to be uniformly distributed in the unit hypersphere, as illustrated in Figure 2b. Therefore, one naive solution is to implement contrastive learning (Chen et al., 2020) to learn representations by distinguishing different views of one example from other examples as follows:

$$\mathcal{L}_{\text{SimCLR}} = - \sum_{z_j \in P \cup U} \log \frac{\text{sim}(z_j, z_j^+)}{\sum_{z_k \in P \cup U} \text{sim}(z_j, z_k)}, \tag{2}$$

where $z_j = g_e(x_j)$ is the hidden representation, $U, P$ are slightly abused to denote the empirical distributions of the hidden representations of the unprotected and protected groups, $z_j^+$ is obtained by an augmentation function to form a positive pair with $z_j$, and $\text{sim}(a, b) = \exp(\frac{a^T b}{|a||b|})$. By minimizing $\mathcal{L}_{\text{SimCLR}}$, we encourage the uniformity of the representations of the two groups.

However, as shown in Figure 2b, although the protected examples deviate from anomalies after encouraging uniformity, group fairness could not be guaranteed by the traditional contrastive learning loss. To promote fairness between the protected group and the unprotected group, we further propose to maximize the cosine similarity between the representations of the protected and unprotected

group, as shown in Figure 2c. Formally, we minimize the following fairness-aware contrastive loss:

$$\mathcal{L}_{\text{FAC}} = -\log \frac{\frac{1}{mn} \sum_{j\in[n]} \sum_{k\in[m]} \text{sim}\left(z_j^P, z_k^U\right)}{\frac{1}{m(m-1)} \sum_{j\neq k} \text{sim}\left(z_j^U, z_k^U\right) + \frac{1}{n(n-1)} \sum_{j\neq k} \text{sim}\left(z_j^P, z_k^P\right)} \tag{3}$$

$$= \underbrace{-\log \left(\frac{\sum_j \sum_k \text{sim}\left(z_j^P, z_k^U\right)}{mn}\right)}_{\mathcal{L}_{\text{fair}}} + \underbrace{\log \left(\frac{\sum_{j\neq k} \text{sim}\left(z_j^U, z_k^U\right)}{m(m-1)} + \frac{\sum_{j\neq k} \text{sim}\left(z_j^P, z_k^P\right)}{n(n-1)}\right)}_{\mathcal{L}_{\text{unif}}}$$

Following the interpretation of contrastive loss in Wang & Isola (2020), the numerator (i.e., $\mathcal{L}_{\text{fair}}$) can be interpreted as ensuring the fairness of two groups by maximizing the representation similarity between different groups, and the denominator (i.e., $\mathcal{L}_{\text{unif}}$) can be interpreted as encouraging the diversity or uniformity of the representations in the unit hypersphere within each group. Besides, we show that our proposed fair contrastive regularization term guarantees group fairness with theoretical support in Section 4.

## 3.2 RE-BALANCING AUTOENCODER

We then introduce the autoencoder-based module of our method. The existing autoencoder-based AD frameworks (Song et al., 2021; Audibert et al., 2020) aim to optimize the following reconstruction loss:

$$\mathcal{L}_{\text{REC}} = \sum_{x_i \in P \cup U} \|x_i - G(x_i)\|^2 = \underbrace{\sum_{i=1}^{n} \|x_i^P - G\left(x_i^P\right)\|^2}_{\mathcal{L}_P} + \underbrace{\sum_{i=1}^{m} \|x_i^U - G\left(x_i^U\right)\|^2}_{\mathcal{L}_U}. \tag{4}$$

As these AD approaches fail to consider the data imbalance nature of the protected and unprotected groups, the learning objective in Equation (4) tends to solely concentrate on learning frequent patterns of the unprotected group (i.e., $\mathcal{L}_U$), yielding higher reconstruction errors for the examples from the protected group. Consequently, existing methods usually overly flag the examples from the protected group as anomalies, thus having a higher recall difference, as illustrated in Figure 1.

To address the data imbalance issue between the two groups (i.e., **C1** in the introduction), we design a re-balancing autoencoder by minimizing the reweighted reconstruction loss as follows:

$$\mathcal{L}_{\text{REC}} = (1-\epsilon)\mathcal{L}_U + \epsilon\mathcal{L}_P, \tag{5}$$

A proper weight $\epsilon$ should promote the model fitting on the normal examples in both protected and unprotected groups. Consider the four subgroups of data samples in the task of fair anomaly detection: unprotected/protected normal examples (UN/PN) and unprotected/protected anomalies (UA/PA). Since ideally the model should only fit UN and PN, we assume that the model is capable of fitting two out of the four subgroups. For the design of $\epsilon$ we have the following lemma:

**Lemma 3.1.** *Let $\mathcal{L}_0^t$ denote the loss of the unfitted model on the subgroup $t \in \{UN, PN, UA, PA\}$, and let $\mathcal{L}_1^t$ denote the loss of the fitted model on the subgroup $t$. $\Delta^t = \mathcal{L}_0^t - \mathcal{L}_1^t > 0$ means the difference of loss between the fitted model and the unfitted one on the subgroup $t$. A proper weight $\epsilon$ for model fitting on normal examples in both protected and unprotected groups should be within the range $\frac{\Delta^{UA}}{\Delta^{UA}+\Delta^{PN}} < \epsilon < \frac{\Delta^{UN}}{\Delta^{UN}+\Delta^{PA}}$ such that fitting normal samples of both groups leads to a lower loss compared to fitting abnormal samples from either group.*

Although $\frac{\Delta^{UA}}{\Delta^{UA}+\Delta^{PN}}$ and $\frac{\Delta^{UN}}{\Delta^{UN}+\Delta^{PA}}$ are unknown, we propose a design of $\epsilon$ that provably lies in this range: $\epsilon = \frac{\mathcal{L}_0^U - \mathcal{L}_U}{\mathcal{L}_0^U - \mathcal{L}_U + \mathcal{L}_0^P - \mathcal{L}_P}$ where $\mathcal{L}_0^U = \mathcal{L}_0^{UN} + \mathcal{L}_0^{UA}$ and $\mathcal{L}_0^P = \mathcal{L}_0^{PN} + \mathcal{L}_0^{PA}$. We estimate $\mathcal{L}_0^U = \sum_{i\in U} \|x_i - \overline{G_U(x)}\|^2$ where $\overline{G_U(x)} = \frac{1}{|U|} \sum_{i\in U} G(x_i)$, and $\mathcal{L}_0^P = \sum_{i\in P} \|x_i - \overline{G_P(x)}\|^2$ where $\overline{G_P(x)} = \frac{1}{|P|} \sum_{i\in P} G(x_i)$. The proof of Lemma 3.1 and the justification of our design are provided in Appendix E.1. Finally, the overall training scheme of FADIG is to minimize:

$$\mathcal{L}_{\text{overall}} = \mathcal{L}_{\text{REC}} + \alpha\mathcal{L}_{\text{FAC}},$$

where $\alpha$ is a hyperparameter to balance the reconstruction loss and the contrastive loss. During the inference stage, we rank the reconstruction error of each example and pick the top $k$ examples as anomalies.

## 4 THEORETICAL ANALYSIS

In this section, we show how our proposed method promotes fairness. We focus on the group fairness notions where the difference in certain performance metrics between the two groups is considered. We first introduce the definition of $f$-divergence to help formulate an upper bound on the performance difference of FADIG:

**Definition 4.1.** ($f$-divergence (Ali & Silvey, 1966) ) Let $P$ and $Q$ be two distribution functions with densities $p$ and $q$, respectively. Let $p$ be absolutely continuous w.r.t $q$ and both be absolutely continuous with respect to a base measure $dx$. Let $f : \mathbb{R}_+ \to \mathbb{R}$ be a convex, lower semi-continuous function that satisfies $f(1) = 0$. The $f$-divergence $D_f$ is defined as:

$$D_f(P \parallel Q) = \int q(x) f\left(\frac{p(x)}{q(x)}\right) dx. \tag{6}$$

Many popular divergences that are heavily used in machine learning are special cases of $f$-divergences, and we include some in Table 8 in Appendix D. Nguyen et al. (2010) derived a general variational approach for estimating $f$-divergence from examples by transforming the estimation problem into a variational optimization problem. They show that any $f$-divergence can be written as:

$$D_f(P \parallel Q) \geq \sup_{T \in \mathcal{T}} \mathbb{E}_{x \sim P}[T(x)] - \mathbb{E}_{x \sim Q}[f^*(T(x))] \tag{7}$$

where $f^*$ is the (Fenchel) conjugate function of $f$ defined as $f^*(y) := \sup_{x \in \mathbb{R}_+}\{xy - f(x)\}$, $T : \mathcal{X} \to \text{dom } f^*$, and $\mathcal{T}$ is the set of all measurable functions.

Given that $D_f(P \parallel Q)$ involves the supremum over all measurable functions and does not account for the hypothesis class, and that it cannot be estimated from finite examples of arbitrary distributions (Kifer et al., 2004), we further consider a discrepancy which helps relieve these issues based on the variational characterization of $f$-divergence in Equation (7):

**Definition 4.2.** ($D_{h,\mathcal{H}}^f$ discrepancy (Acuna et al., 2021)) Let $f^*$ be the Fenchel conjugate of a convex, lower semi-continuous function $f$ that satisfies $f(1) = 0$, and let $\hat{T}$ be a set of measurable functions such that $\hat{T} = \{\ell(h(x), h'(x)) : h, h' \in \mathcal{H}\}$ where $\ell$ is a loss function and $\mathcal{H}$ is the hypothesis space. We define the discrepancy between the two distributions $P$ and $Q$ as:

$$D_{h,\mathcal{H}}^f(P \parallel Q) := \sup_{h' \in \mathcal{H}} |\mathbb{E}_{x \sim P}[\ell(h(x), h'(x))] - \mathbb{E}_{x \sim Q}[f^*(\ell(h(x), h'(x)))]|$$

From the definition we can easily get $D_{h,\mathcal{H}}^f(P \parallel Q) \leq D_f(P \parallel Q)$. Next we introduce a useful tool, Rademacher complexity (Shalev-Shwartz & Ben-David, 2014) (detailed definition provided in Appendix C). Recall that we previously defined $R_{\mathcal{D}}^\ell(h) := R_{\mathcal{D}}^\ell(h, a) = \mathbb{E}_{x \sim \mathcal{D}}[\ell(h(x), a(x))]$. We introduce a commonly used property of Rademacher complexity:

**Lemma 4.3.** (*Property of Rademacher complexity (Mohri et al., 2018)). For any $\delta \in (0,1)$, with probability at least $1 - \delta$ over the draw of an i.i.d. samples $D$ of size $|D|$, the following inequality holds for all $h \in \mathcal{H}$:*

$$|R_D^\ell(h) - \hat{R}_D^\ell(h)| \leq 2\mathfrak{R}_D(\ell \circ \mathcal{H}) + \sqrt{\frac{\log \frac{1}{\delta}}{2|D|}} \tag{8}$$

where $\mathfrak{R}_D(\ell \circ \mathcal{H})$ is the Rademacher complexity of the function class $\ell \circ \mathcal{H}$ given data $D$. With this property, we now show that $D_{h,\mathcal{H}}^f$ can be estimated from finite examples:

**Lemma 4.4.** *Suppose $\ell : \mathcal{Y} \times \mathcal{Y} \to [0,1]$, $f^*$ is $L$-Lipschitz continuous, and $[0,1] \subseteq \text{dom } f^*$. Let $U$ and $P$ be two empirical distributions corresponding to datasets containing $m$ and $n$ data points sampled i.i.d. from $P_U$ and $P_P$, respectively. Let us denote $\mathfrak{R}$ as the Rademacher complexity of a given hypothesis class, and define $\ell \circ \mathcal{H} := \{x \mapsto \ell(h(x), h'(x)) : h, h' \in \mathcal{H}\}$. For any $\delta \in (0,1)$, with probability at least $1 - \delta$, we have:*

$$|D_{h,\mathcal{H}}^f(P_U \| P_P) - D_{h,\mathcal{H}}^f(U \| P)| \leq 2\mathfrak{R}_{P_U}(\ell \circ \mathcal{H}) + 2L\mathfrak{R}_{P_P}(\ell \circ \mathcal{H}) + \sqrt{\frac{\log \frac{1}{\delta}}{2n}} + \sqrt{\frac{\log \frac{1}{\delta}}{2m}} \tag{9}$$

Table 2: Characteristics of datasets.

| Datasets | Unprotected Group | | Protected Group | | #Features | Sensitive Attribute | Anomaly Definition |
|---|---|---|---|---|---|---|---|
| | #Instances | #Anomaly | #Instances | #Anomaly | | | |
| MNIST-USPS | 7,785 | 882 | 1,876 | 323 | 1,024 | Source of the digits | Digit 0 or not |
| MNIST-Invert | 7,344 | 441 | 408 | 38 | 1,024 | Color of the digits | Digit 0 or not |
| COMPAS | 1,839 | 325 | 299 | 39 | 8 | Race | Reoffending or not |
| CelebA | 41,919 | 4,008 | 7,300 | 1,142 | 39 | Gender | Attractive or not |

Lemma 4.4 shows that the empirical discrepancy $D_{h,\mathcal{H}}^f$ converges to the true discrepancy, and the gap is bounded by the complexity of the hypothesis class and the number of examples.

## 4.1 FAIRNESS BOUNDS

We now provide a fairness bound to estimate the performance difference between the protected and unprotected groups using the previously defined $D_{h,\mathcal{H}}^f$ divergence.

**Theorem 4.5.** *Let $h^*$ be the ideal joint hypothesis, i.e., $h^* = \arg\min_{h \in \mathcal{H}} R_U^\ell(h) + R_P^\ell(h)$. The risk difference between the two groups is upper bounded by:*

$$R_P^\ell(h) - R_U^\ell(h) \leq D_{h,\mathcal{H}}^f(P_U \| P_P) + R_U^\ell(h^*) + R_P^\ell(h^*). \tag{10}$$

For the upper bound on the RHS, the first term corresponds to the discrepancy between the marginal distributions, and the remaining two terms measure the risk of the ideal joint hypothesis. If $\mathcal{H}$ is expressive enough and the labeling functions of the protected and unprotected groups are similar, the last two terms could be reduced to a small value.

**Theorem 4.6.** *(Fairness with Rademacher Complexity) Under the same conditions as in Lemma 4.4, for any $\delta \in (0,1)$, with probability at least $1 - \delta$, we have:*

$$R_P^\ell(h) - R_U^\ell(h) \leq D_f(U \| P) + \hat{R}_U^\ell(h^*) + \hat{R}_P^\ell(h^*)$$
$$+ 4\mathfrak{R}_U(\ell \circ \mathcal{H}) + 2(L+1)\mathfrak{R}_P(\ell \circ \mathcal{H}) + 2\sqrt{\frac{\log \frac{1}{\delta}}{2m}} + 2\sqrt{\frac{\log \frac{1}{\delta}}{2n}} \tag{11}$$

Under the assumption of an ideal joint hypothesis, fairness (i.e., the risk difference between the protected and unprotected groups) can be improved by minimizing the discrepancy between the hidden representation of the samples from two groups and regularizing the model to limit the complexity of the hypothesis class. The detailed proofs of the lemma and the theorems are in Appendix E.2 and E.3. We further motivate why minimizing the objective $\mathcal{L}_{\text{FAC}}$ leads to small $D_f(U \| P)$ for total variation in Appendix E.4.

## 5 EXPERIMENTS

In this section, we experimentally analyze and compare our proposed FADIG with other anomaly detection methods. We try to answer the following research questions:

- RQ1: How does FADIG compare with other baselines on imbalanced datasets?
- RQ2: How does FADIG perform with different ratios of the two groups?
- RQ3: How does each module contribute to FADIG?

### 5.1 EXPERIMENTAL SETUP

**Datasets:** We conduct experiments on two image datasets, MNIST-USPS and MNIST-Invert (Zhang & Davidson, 2021), and two tabular datasets, COMPAS (Angwin et al., 2022) and CelebA (Liu et al., 2015). The characteristics of the datasets are presented in Table 2.

**Baseline Methods:** In our experiments, we compare our proposed framework FADIG with the following fairness-aware anomaly detection baselines: (1) **FairOD** (Shekhar et al., 2021), a fair

Table 3: Performance on image datasets. The best score is marked in bold.

| Methods | MNIST-USPS (K=1200) | | | | MNIST-Invert (K=500) | | | |
|---|---|---|---|---|---|---|---|---|
| | Recall@K | ROCAUC | Rec Diff | Time(s) | Recall@K | ROCAUC | Rec Diff | Time(s) |
| FairOD | 12.35±1.13 | 50.00±0.28 | 11.56±0.64 | 29.57 | 7.52±0.74 | 50.40±0.20 | 8.26±1.27 | 20.25 |
| DCFOD | 12.63±0.33 | 50.09±0.27 | 8.99±0.83 | 710.33 | 6.95±0.91 | 50.54±0.54 | **7.23±2.02** | 1277.31 |
| FairSVDD | 15.62±1.52 | 58.33±1.18 | 13.75±2.56 | 768.79 | 12.41±0.76 | 49.67±3.98 | 12.46±2.12 | 843.12 |
| MCM | 39.75±0.23 | 78.80±1.02 | 55.81±0.80 | 417.09 | 25.35±0.56 | 80.96±0.49 | 80.13±1.41 | 752.36 |
| NSNMF | 39.16±0.84 | 65.38±0.58 | 62.90±3.84 | 28.53 | 51.79±0.61 | 74.21±0.34 | 51.07±1.79 | 18.97 |
| Recontrast | 64.29±3.18 | 83.46±3.77 | 41.16±5.63 | 116.75 | 64.22±1.60 | 85.13±5.19 | 56.50±11.23 | 117.15 |
| FADIG | **67.19±0.33** | **91.28±0.46** | **3.77±2.18** | 121.97 | **71.82±0.63** | **97.99±0.07** | 9.78±3.10 | 60.42 |

Table 4: Performance on tabular datasets. The best score is marked in bold.

| Methods | COMPAS (K=350) | | | | CelebA (K=5000) | | | |
|---|---|---|---|---|---|---|---|---|
| | Recall@K | ROCAUC | Rec Diff | Time(s) | Recall@K | ROCAUC | Rec Diff | Time(s) |
| FairOD | 16.56±2.12 | 50.09±1.28 | 7.97±1.23 | 4.18 | 8.93±0.14 | 49.94±0.12 | **0.68±0.56** | 78.92 |
| DCFOD | 16.08±1.94 | 49.55±1.21 | 9.81±1.76 | 115.86 | 9.66±0.69 | 49.92±0.14 | 7.83±1.26 | 2517.68 |
| FairSVDD | 15.33±2.10 | 52.68±5.29 | 11.57±4.06 | 6.81 | 10.19±0.50 | 58.40±1.02 | 10.95±1.93 | 243.17 |
| MCM | 21.10±0.54 | 50.97±0.43 | 6.29±2.66 | 38.12 | 11.03±0.38 | 46.23±3.46 | 26.15±9.31 | 640.12 |
| NSNMF | 22.92±0.32 | 57.97±0.66 | 36.78±1.71 | 7.69 | 10.91±0.54 | 50.45±0.30 | 8.04±1.33 | 1927.55 |
| FADIG | **34.38±0.36** | **61.45±0.47** | **5.97±4.34** | 19.88 | **11.96±0.49** | **59.43±0.42** | 4.72±1.26 | 48.93 |

AD method which incorporates various group fairness criteria including flag rate parity, statistical parity and group fidelity into its training; (2) **DCFOD** (Song et al., 2021), a fair deep clustering-based method, which leverages deep clustering to discover the intrinsic cluster structure and out-of-structure instances; (3) **FairSVDD** (Zhang & Davidson, 2021), an adversarial network to de-correlate the relationships between sensitive attributes and the learned representations. We also compare with the following fairness-agnostic AD baselines: (4) **MCM** (Yin et al., 2024), a masked modeling method to address AD by capturing intrinsic correlations between features in the training set; (5) **NSNMF** (Ahmed et al., 2021), a non-negative matrix factorization method, which incorporates the neighborhood structural similarity information to improve the anomaly detection performance; (6) **ReContrast** (Guo et al., 2023), a reconstructive contrastive learning-based method for domain-specific anomaly detection. Notice that as ReContrast is designed for image data, we only evaluate it on MNIST-USPS and MNIST-Invert datasets.

**Metrics:** To measure the model performance and group fairness, we choose three widely-used metrics (Shekhar et al., 2021; Zhang & Davidson, 2021; Ahmed et al., 2021): (1) **Recall@K**, which measures the proportion of anomalies found in the top-k recommendations; (2) **ROCAUC**, which computes the area under the receiver operating characteristic curve; (3) **Rec Diff**, which measures the absolute value of the recall difference between two groups.

**Training details:** For the COMPAS dataset, we use a two-layer MLP with hidden units of [32, 32]. For all the other datasets, we use MLP with one hidden layer of dimension 128. We set the hyperparameter $\alpha = 4$ across all the data sets. For $\epsilon = \frac{\mathcal{L}_0^U - \mathcal{L}_U}{\mathcal{L}_0^U - \mathcal{L}_U + \mathcal{L}_0^P - \mathcal{L}_P}$, we estimate $\mathcal{L}_0^U = \sum_{i \in U} \|x_i - \overline{G_U(x)}\|^2$ where $\overline{G_U(x)} = \frac{1}{|U|} \sum_{i \in U} G(x_i)$, and $\mathcal{L}_0^P = \sum_{i \in P} \|x_i - \overline{G_P(x)}\|^2$ where $\overline{G_P(x)} = \frac{1}{|P|} \sum_{i \in P} G(x_i)$. We include the results with different choices of $\alpha$ in Appendix F.4 and different designs of $\mathcal{L}_0^U$ and $\mathcal{L}_0^P$ for $\epsilon$ in Appendix E.1. All our experiments were executed using one Tesla V100 SXM2 GPUs, supported by a 12-core CPU operating at 2.2GHz. We provide more implementation details in Appendix F.1.

## 5.2 EFFECTIVENESS AND EFFICIENCY OF FADIG (RQ1)

We first evaluate the effectiveness and efficiency of FADIG through comparison with baselines across four datasets by four independent runs. The task performance (*i.e.*, Recall@$K$ and ROCAUC), group fairness measure (*i.e.*, Rec Diff), and their average training time are presented in Tables 3 and 4 (see Appendix F.2 for results with different $K$). We can observe that the fair AD baselines (FairOD, DCFOD, and FairSVDD) typically exhibit low discrepancies in recall. However,

Table 5: Performance on MNIST-USPS with different ratios. The best score is marked in bold.

| Methods | $|U| : |P| = 1 : 1$ (K=650) | | | $|U| : |P| = 2 : 1$ (K=1000) | | | $|U| : |P| = 4 : 1$ (K=1200) | | |
|---|---|---|---|---|---|---|---|---|---|
| | Recall@K | ROCAUC | Rec Diff | Recall@K | ROCAUC | Rec Diff | Recall@K | ROCAUC | Rec Diff |
| FairOD | 17.52±1.17 | 50.13±0.64 | 2.14±0.62 | 17.30±1.24 | 49.73±0.74 | 5.11±0.55 | 13.61±0.22 | 50.22±0.13 | 10.58±1.01 |
| DCFOD | 17.08±0.50 | 50.09±0.30 | 3.25±0.94 | 16.92±0.81 | 49.54±0.42 | 2.76±0.51 | 14.14±1.03 | 50.44±0.60 | 7.11±0.83 |
| FairSVDD | 24.56±2.95 | 54.87±3.36 | 14.24±7.90 | 18.09±3.46 | 52.77±1.72 | 4.85±3.75 | 21.10±2.79 | 63.46±9.56 | 18.38±4.91 |
| MCM | 52.22±1.35 | 74.62±1.24 | 17.13±2.73 | 53.63±1.76 | 76.80±1.04 | 8.17±6.36 | 41.99±4.06 | 74.09±0.45 | 22.85±4.60 |
| NSNMF | 48.71±0.39 | 68.96±0.24 | 40.25±2.17 | 41.07±2.77 | 64.08±1.67 | 54.18±3.11 | 38.87±1.09 | 64.71±0.63 | 62.98±1.47 |
| Recontrast | 45.92±1.85 | 80.17±3.08 | 42.52±3.31 | 51.39±1.75 | 83.13±2.94 | 26.16±1.79 | 57.69±2.36 | 79.17±4.09 | 20.69±3.57 |
| FADIG | **65.58±0.47** | **85.38±0.37** | **0.93±0.87** | **66.84±0.83** | **89.17±0.09** | **2.32±1.08** | **66.63±0.72** | **90.15±0.22** | **1.84±0.68** |

Table 6: Performance on COMPAS dataset with different ratios. The best score is marked in bold.

| Methods | $|U| : |P| = 1 : 1$ (K=80) | | | $|U| : |P| = 2 : 1$ (K=120) | | | $|U| : |P| = 5 : 1$ (K=240) | | |
|---|---|---|---|---|---|---|---|---|---|
| | Recall@K | ROCAUC | Rec Diff | Recall@K | ROCAUC | Rec Diff | Recall@K | ROCAUC | Rec Diff |
| FairOD | 13.68±2.67 | 50.10±0.85 | 11.97±1.48 | 13.11±0.50 | 50.11±0.74 | 6.60±0.97 | 12.54±1.37 | 49.58±0.87 | 7.68±0.72 |
| DCFOD | 11.54±4.62 | 48.50±2.69 | 7.69±4.445 | 15.95±3.00 | 53.28±0.75 | 10.68±2.67 | 12.96±2.02 | 49.76±1.16 | 6.36±0.70 |
| FairSVDD | 16.24±2.18 | 52.34±1.38 | 6.84±3.20 | 14.53±1.84 | 51.69±2.15 | 7.69±3.77 | 14.10±4.53 | 50.04±4.98 | 14.87±7.54 |
| MCM | 18.38±0.60 | 40.77±0.25 | 7.69±3.63 | 16.24±0.01 | 40.42±0.12 | 10.26±4.80 | 18.81±0.60 | 44.04±0.15 | 5.76±2.31 |
| NSNMF | 20.08±0.74 | 53.86±0.42 | 14.53±10.36 | 19.09±1.31 | 53.28±0.75 | 10.68±2.67 | 20.09±2.22 | 53.86±1.28 | 10.77±5.40 |
| FADIG | **29.91±0.74** | **61.87±1.89** | **3.42±1.48** | **28.42±0.43** | **57.39±2.84** | **1.92±1.72** | **29.77±1.31** | **58.05±1.34** | **4.83±0.78** |

they also tend to suffer from reduced recall rates and ROCAUC scores, suggesting a compromise in overall task performance to enhance fairness. On the other hand, the baselines that do not account for fairness, including MCM, NSNMF, and ReContrast, demonstrate high recall rates and ROCAUC scores but often at the expense of fairness, as evidenced by significant disparities across groups (*i.e.*, a higher Rec Diff). Our FADIG instead addresses the challenge of imbalance between the groups and the imbalanced distributions of normal examples and anomalies. Remarkably, FADIG not only excels in task performance but also elevates the level of fairness, underscoring the effectiveness of our design in harmonizing fairness with anomaly detection in scenarios characterized by data imbalance. On the other hand, the training time of FADIG is always among the top 4 fastest methods across different datasets, showing the efficiency of our method.

## 5.3 DATA IMBALANCE STUDY (RQ2)

To further study the performance of FADIG in handling imbalanced data, we vary the levels of group imbalance within the image dataset MNIST-USPS and the tabular dataset COMPAS. We report the average results of four independent runs in Tables 5 and 6. The tables demonstrate that FADIG consistently outperforms the baselines in terms of both task efficacy and fairness across different group ratios. The advantages of using FADIG become more pronounced with increasing level of group imbalance. For instance, while the performance of fair AD baselines drops with higher imbalance ratios on the MNIST-USPS dataset, FADIG adeptly sustains superior task performance alongside enhanced fairness levels, showcasing its robustness against data imbalance.

## 5.4 ABLATION STUDY (RQ3)

To validate the necessity of each module in FADIG, we conduct an ablation study to demonstrate the necessity of each component of FADIG on the MNIST-USPS and COMPAS datasets. The experimental results are presented in Figures 3 and 4, where (a) and (b) show the recall rate and recall difference, respectively. Specifically, FADIG-R refers to a variant of our method replacing the re-balancing autoencoder with $\mathcal{L}_{\text{REC}}$ in Equation (4); FADIG-N and FADIG-D remove $\mathcal{L}_{\text{fair}}$ and $\mathcal{L}_{\text{unif}}$ in Equation (3), respectively; FADIG-C substitutes the proposed fair contrastive loss with the traditional contrastive loss (*i.e.*, $\mathcal{L}_{\text{SimCLR}}$). We have the following observations: (1) FADIG greatly outperforms FADIG-N and FADIG-D, which suggests that $\mathcal{L}_{\text{fair}}$ and $\mathcal{L}_{\text{unif}}$ are two essential components in our designed method. (2) FADIG-C sometimes has the competitive performance as FADIG with respect to the recall rate, but it always has a large recall difference. This suggests that without proper regularization for representation similarity between the two groups, the model will exhibit unfair behaviors. Different from FADIG-C, FADIG achieves a much lower recall dif-

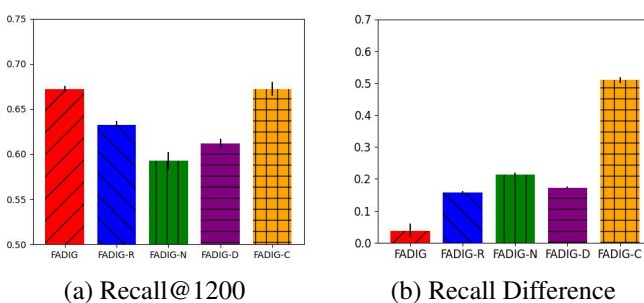

(a) Recall@1200      (b) Recall Difference

Figure 3: Ablation Study on MNIST-USPS dataset.

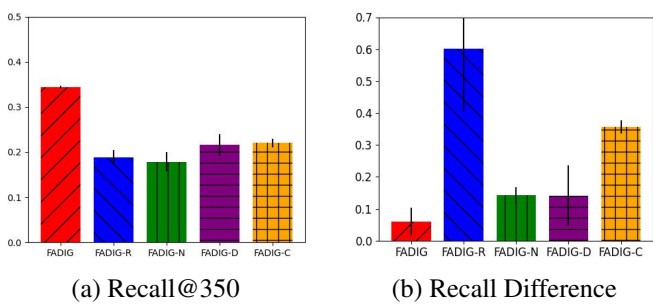

(a) Recall@350      (b) Recall Difference

Figure 4: Ablation Study on COMPAS dataset.

ference, which verifies our theoretical analysis that our proposed method could guarantee group fairness. (3) Compared with FADIG, FADIG-R has a lower recall rate and a higher recall difference. This indicates that replacing the re-balancing autoencoder with classical $\mathcal{L}_{\text{REC}}$ in Equation (4) results in worse performance, which verifies our conjecture that the traditional learning objective tends to mainly focus on learning the frequent patterns of the unprotected group while ignoring the protected group.

We include the parameter analysis in Appendix F.4 and find that FADIG is robust to the choice of $\alpha$. We compare our method with other reweighting heuristics in Appendix F.3, test it on different anomaly types in Appendix F.5, and compare with more baselines (Appendix F.6) on more tasks (Appendix F.7).

## 6 CONCLUSION

In this paper, we introduce FADIG, a fairness-aware anomaly detection method, designed for handling the imbalanced data scenario in the context of anomaly detection. Specifically, FADIG maximizes the similarity between the protected and unprotected groups to ensure fairness through the fairness-aware contrastive learning based module. To address the negative impact of imbalanced data, the re-balancing autoencoder module is proposed to automatically reweight the importance of both the protected and unprotected groups with the learnable weight. Theoretically, we provide the upper bound with Rademacher complexity for the discrepancy between two groups and ensure group fairness through the proposed contrastive learning regularization. Empirical studies demonstrate the effectiveness and efficiency of FADIG across multiple real-world datasets.

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

## A    NOTATIONS

Table 7: Notation Table.

| Symbol | Description |
|---|---|
| $x$ | input feature |
| $\mathcal{P}_P$ | Protected group's distribution |
| $\mathcal{P}_U$ | Unprotected group's distribution |
| $P$ | Protected group's empirical distribution |
| $U$ | Unprotected group's empirical distribution |
| $n/m$ | Size of protected/unprotected group |
| $a_P/a_U$ | labeling functions on protected/unprotected group |
| $\ell$ | Loss function |
| $R_D^\ell(h)$ | Risk of hypothesis $h$ over data $D$ |
| $\hat{R}_D^\ell(h)$ | Empirical risk of hypothesis $h$ over data $D$ |
| $s(x)$ | Anomaly score of example $x$ |
| $\mathfrak{R}_D(\mathcal{F})$ | Rademacher complexity of $\mathcal{F}$ given data $D$ |
| $D_f(P \parallel Q)$ | f-divergence between distributions $P$ and $Q$ |

## B    RELATED WORK

**Unsupervised Anomaly Detection.** Anomaly detection has been widely studied for decades in many real-world applications, including fraud detection in the finance domain (West & Bhattacharya, 2016; Huang et al., 2018), pathology analysis in the medical domain (Faust et al., 2018; Shvetsova et al., 2021), intrusion detection for cyber-security (Liao et al., 2013; Ahmad et al., 2021), and fault detection in safety-critical systems (Ju et al., 2021), etc. Given various types of anomalies (Breunig et al., 2000; Kriegel et al., 2009; Bouman et al., 2024), the authors of Pang et al. (2022) divide the existing anomaly detection methods into two major branches. The methods (Audibert et al., 2020; Chen et al., 2021; Hou et al., 2021; Yan et al., 2021; Wang et al., 2023) in the first branch aim to learn the patterns for the normal samples by a feature extractor. For instance, Audibert et al. (2020) is an encoder-decoder anomaly detection method, which learns how to amplify the reconstruction error of anomalies with adversarial training; Chen et al. (2021) proposes a GAN-based autoencoder model to learn the normal pattern of multivariate time series, and detect anomalies by selecting the samples with the higher reconstruction error. Our work also uses the autoencoder model. Compared with the works in the first branch, we design an auto-reweighted training of the reconstruction errors, and mitigate the representation disparity between groups with contrastive learning-based regularization. The second branch aims at learning scalar anomaly scores in an end-to-end fashion (Sohn et al., 2021; Li et al., 2023; Jiang et al., 2022). Notably, the authors of Sohn et al. (2021) combine distribution-augmented contrastive regularization with a one-class classifier to detect anomalies. While Sohn et al. (2021) uses image augmentations, such as rotations, to form positive pairs and negative pairs for contrastive learning, we use existing examples for contrastive learning and thus our method is applicable to various types of data, not limited to image data. To be more specific, we design a fairness-aware contrastive learning loss which minimizes the representation disparity of the groups for fairness, and encourages the uniformity within each group for better anomaly detection.

**Fair Machine Learning.** Fair Machine Learning aims to amend the biased machine learning models to be fair or invariant regarding specific variables. A surge of research in fair machine learning has been done in the machine learning community (Kobren et al., 2019; Zemel et al., 2013; Bolukbasi et al., 2016; Hashimoto et al., 2018; Zhang et al., 2018; Park et al., 2022). For example, Zemel et al. (2013) presents a learning algorithm for fair classification by enforcing group fairness and individual

fairness in the obtained data representation; Bolukbasi et al. (2016) proposes approaches to quantify and reduce bias in word embedding vectors that are trained from real-world data; in Hashimoto et al. (2018), the authors develop a robust optimization framework that minimizes the worst case risk over all distributions and preserves the minority group in an imbalanced data set; in Zhang et al. (2018), the authors present an adversarial-learning based framework for mitigating the undesired bias in modern machine learning models. Park et al. (2022) proposes a fair supervised contrastive loss to train a fair representation model. However, they rely on target labels and need negative samples since their method is based on supervised contrastive learning. Instead, our designed fair contrastive learning loss uses examples from different groups in an unsupervised way to minimize the representation disparity of the groups and encourage the uniformity within each group. In the field of fair anomaly detection, Zhang & Davidson (2021) utilizes the adversarial generative nets to ensure group fairness and use one-class classification to detect the anomalies; Song et al. (2021) introduces fairness adversarial training and proposes a novel dynamic weight to reduce the negative impacts from outlier points. The existing fair anomaly detection methods (Song et al., 2021; Zhang & Davidson, 2021; Fioresi et al., 2023) tend to suffer from the representation disparity issue in the imbalanced data scenario. To address this issue, this paper aims to alleviate the issue of representation disparity in the imbalanced data scenario by introducing the rebalancing autoencoder module and maximizing the uniformity of the samples in the latent space via contrastive learning regularization.

## C  RADEMACHER COMPLEXITY

The Rademacher complexity for a function class is:

**Definition C.1.** (Rademacher Complexity (Shalev-Shwartz & Ben-David, 2014)) Given a space $\mathcal{X}$, and a set of i.i.d. examples $D = \{x_1, x_2, ..., x_{|D|}\} \subseteq \mathcal{X}$, for a function class $\mathcal{F}$ where each function $r : \mathcal{X} \to \mathbb{R}$, the empirical Rademacher complexity of $\mathcal{F}$ is given by:

$$\mathfrak{R}_D(\mathcal{F}) = \mathbb{E}_\sigma \left[ \sup_{r \in \mathcal{F}} \left( \frac{1}{|D|} \sum_{i=1}^{|D|} \sigma_i r(z_i) \right) \right] \tag{12}$$

Here, $\sigma_1, ..., \sigma_m$ are independent random variables uniformly drawn from $\{-1, 1\}$.

## D  DIVERGENCES

We include some popular $f$-divergences in Table 8.

Table 8: Popular $f$-divergences and their conjugate functions.

| Divergence | $f(x)$ | Conjugate $f^*(t)$ | $f'(1)$ | Activation func. |
|---|---|---|---|---|
| Kullback-Leibler (KL) | $x \log x$ | $\exp(t-1)$ | 1 | $x$ |
| Reverse KL (KL-rev) | $-\log x$ | $-1 - \log(-t)$ | -1 | $-\exp x$ |
| Jensen-Shannon (JS) | $-x + 1 \log \frac{1+x}{2} + x \log x$ | $-\log(2 - e^t)$ | 0 | $\log \frac{2}{1+\exp(-x)}$ |
| Pearson $\chi^2$ | $(x-1)^2$ | $\frac{t^2}{4} + t$ | 0 | $x$ |
| Total Variation (TV) | $\frac{1}{2}|x - 1|$ | $1_{-1/2 \leq t \leq 1/2}$ | $[-1/2, 1/2]$ | $\frac{1}{2} \tanh x$ |

## E  PROOFS

### E.1  PROOF OF LEMMA 3.1

Let us divide the data into four types: unprotected normal examples (UN), protected normal examples (PN), unprotected anomalies (UA), and protected anomalies (PA). For type $t \in \{$UN, PN, UA, PA$\}$, let $\mathcal{L}_0^t$ denote the loss of the unfitted model on $t$ and $\mathcal{L}_1^t$ as the loss of the fitted model on $t$, $\Delta^t = \mathcal{L}_0^t - \mathcal{L}_1^t > 0$. Assuming that the model can only fit two sets of data, to ensure that the model fits the sets of protected normal examples and unprotected normal examples, we need the following 5 inequalities to hold:

Table 9: Performance of FADIG with different designs of $\mathcal{L}_0^U$ and $\mathcal{L}_0^P$.

| Methods | MNIST-USPS (K=1200) | | | MNIST-Invert (K=500) | | |
|---|---|---|---|---|---|---|
| | Recall@K | ROCAUC | Rec Diff | Recall@K | ROCAUC | Rec Diff |
| loss1 | 67.16±0.37 | 91.27±0.49 | 3.73±2.13 | 72.37±0.32 | 98.03±0.01 | 6.75±0.34 |
| loss2 | 66.47±1.73 | 90.60±0.52 | 4.78±2.36 | 72.44±0.74 | 98.04±0.03 | 7.22±0.21 |
| loss3 | 66.31±0.65 | 91.37±0.88 | 6.32±1.74 | 71.39±1.96 | 97.22±1.42 | 8.95±0.92 |
| loss4 | 66.56±2.32 | 90.88±1.67 | 2.54±2.11 | 71.92±3.58 | 97.01±1.85 | 8.96±3.23 |

$$(1-\epsilon)(\mathcal{L}_1^{UN} + \mathcal{L}_0^{UA}) + \epsilon(\mathcal{L}_1^{PN} + \mathcal{L}_0^{PA}) <$$

1. $(1-\epsilon)(\mathcal{L}_0^{UN} + \mathcal{L}_1^{UA}) + \epsilon(\mathcal{L}_1^{PN} + \mathcal{L}_0^{PA})$, implied by $\Delta^{UN} > \Delta^{UA}$ which naturally holds;
2. $(1-\epsilon)(\mathcal{L}_1^{UN} + \mathcal{L}_0^{UA}) + \epsilon(\mathcal{L}_0^{PN} + \mathcal{L}_1^{PA})$, implied by $\Delta^{PN} > \Delta^{PA}$ which naturally holds;
3. $(1-\epsilon)(\mathcal{L}_0^{UN} + \mathcal{L}_1^{UA}) + \epsilon(\mathcal{L}_0^{PN} + \mathcal{L}_1^{PA})$, this case is equivalent to case 1 plus case 2;
4. $(1-\epsilon)(\mathcal{L}_1^{UN} + \mathcal{L}_1^{UA}) + \epsilon(\mathcal{L}_0^{PN} + \mathcal{L}_0^{PA})$, we need $\epsilon > \frac{\Delta^{UA}}{\Delta^{UA}+\Delta^{PN}}$;
5. $(1-\epsilon)(\mathcal{L}_0^{UN} + \mathcal{L}_0^{UA}) + \epsilon(\mathcal{L}_1^{PN} + \mathcal{L}_1^{PA})$, we need $\epsilon < \frac{\Delta^{UN}}{\Delta^{UN}+\Delta^{PA}}$.

So we have: $\frac{\Delta^{UA}}{\Delta^{UA}+\Delta^{PN}} < \epsilon < \frac{\Delta^{UN}}{\Delta^{UN}+\Delta^{PA}}$. We design $\epsilon = \frac{\mathcal{L}_0^U - \mathcal{L}_U}{\mathcal{L}_0^U - \mathcal{L}_U + \mathcal{L}_0^P - \mathcal{L}_P}$, and we discuss the following three cases:

- If $\mathcal{L}_U = \mathcal{L}_1^{UN} + \mathcal{L}_0^{UA}, \mathcal{L}_P = \mathcal{L}_1^{PN} + \mathcal{L}_0^{PA}$, then $\epsilon = \frac{\Delta^{UN}}{\Delta^{UN}+\Delta^{PN}}$, which is within the range;
- If $\mathcal{L}_U = \mathcal{L}_1^{UN} + \mathcal{L}_1^{UA}, \mathcal{L}_P = \mathcal{L}_0^{PN} + \mathcal{L}_0^{PA}$, then $\epsilon = 1$, it encourages to fit $\mathcal{L}_P$;
- If $\mathcal{L}_U = \mathcal{L}_0^{UN} + \mathcal{L}_0^{UA}, \mathcal{L}_P = \mathcal{L}_1^{PN} + \mathcal{L}_1^{PA}$, then $\epsilon = 0$, it encourages to fit $\mathcal{L}_U$.

We estimate $\mathcal{L}_0^U = \sum_{i \in U} \|x_i - \overline{G(x)}\|^2$ where $\overline{G(x)} = \frac{1}{|U|} \sum_{i \in U} G(x_i)$, and $\mathcal{L}_0^P = \sum_{i \in P} \|x_i - \overline{G(x)}\|^2$ where $\overline{G(x)} = \frac{1}{|P|} \sum_{i \in P} G(x_i)$. Let us denote this as loss1. We also provide results on real-world datasets with different designs of estimation in Table 9:

- loss2: $\mathcal{L}_0^U = \sum_{i \in U} \|x_i\|^2$ and $\mathcal{L}_0^P = \sum_{i \in P} \|x_i\|^2$
- loss3: $\mathcal{L}_0^U = \sum_{i \in U} \|G(x_i) - \overline{x}\|^2$ and $\mathcal{L}_0^P = \sum_{i \in P} \|G(x_i) - \overline{x}\|^2$
- loss4: $\mathcal{L}_0^U = \sum_{i \in U} \|x_i - \overline{x}\|^2$ and $\mathcal{L}_0^P = \sum_{i \in P} \|x_i - \overline{x}\|^2$

And we can see that although the results may vary with different estimation designs, our method always performs better than the baselines in both task performance and fairness.

### E.2 PROOF OF LEMMA 4.4

$$D_{h,\mathcal{H}}^f(P_U \| P_P) - D_{h,\mathcal{H}}^f(U\|P) = \sup_{h' \in \mathcal{H}} \{|R_U^\ell(h, h') - R_P^{f^* \circ \ell}(h, h')|\}$$
$$- \sup_{h' \in \mathcal{H}} \{|\hat{R}_U^\ell(h, h') - \hat{R}_P^{f^* \circ \ell}(h, h')|\}$$
$$\leq \sup_{h' \in \mathcal{H}} \|R_U^\ell(h, h') - R_P^{f^* \circ \ell}(h, h')| - |\hat{R}_U^\ell(h, h') - \hat{R}_P^{f^* \circ \ell}(h, h')\|$$
$$\leq \sup_{h' \in \mathcal{H}} |R_U^\ell(h, h') - R_P^{f^* \circ \ell}(h, h') - \hat{R}_U^\ell(h, h') + \hat{R}_P^{f^* \circ \ell}(h, h')|$$
$$= \sup_{h' \in \mathcal{H}} |R_U^\ell(h, h') - \hat{R}_U^\ell(h, h')| + |R_P^{f^* \circ \ell}(h, h') - \hat{R}_P^{f^* \circ \ell}(h, h')|$$
$$\leq 2\mathfrak{R}_{P_U}(\ell \circ \mathcal{H}) + \sqrt{\frac{\log \frac{1}{\delta}}{2m}} + 2\mathfrak{R}_{P_P}(f^* \circ \ell \circ \mathcal{H}) + \sqrt{\frac{\log \frac{1}{\delta}}{2n}}$$

where the last inequality comes from the property of Rademacher complexity. Similarly, by Lemma 5.7 and Definition 3.2 of Mohri et al. (2018) we have: $\mathfrak{R}_{P_P}(f^* \circ \ell \circ \mathcal{H}) \leq L\mathfrak{R}_{P_P}(\ell \circ \mathcal{H})$, with $f^* \circ \ell \circ \mathcal{H} := \{x \mapsto \phi(\ell(h(x), h'(x))) : h, h' \in \mathcal{H}\}$.

### E.3 PROOF OF THEOREM 4.5

First, notice that by definition, $f^*(t) = \sup_{x \in \text{dom} f}(xt - f(x)) \geq t - f(1) = t$. Then we can prove:

$$
\begin{aligned}
R_P^\ell(h, a_P) &\leq R_P^\ell(h, h^*) + R_P^\ell(h^*, a_P) && \text{(triangle inequality } \ell) \\
&= R_P^\ell(h, h^*) + R_P^\ell(h^*, a_P) - R_U^\ell(h, h^*) + R_U^\ell(h, h^*) \\
&\leq R_P^{f^* \circ \ell}(h, h^*) - R_U^\ell(h, h^*) + R_U^\ell(h, h^*) + R_P^\ell(h^*, a_P) \\
&\leq |R_P^{f^* \circ \ell}(h, h^*) - R_U^\ell(h, h^*)| + R_U^\ell(h, h^*) + R_P^\ell(h^*, a_P) \\
&\leq D_{h,\mathcal{H}}^f(P_U \| P_P) + R_U^\ell(h, h^*) + R_P^\ell(h^*, a_P) \\
&\leq D_{h,\mathcal{H}}^f(P_U \| P_P) + R_U^\ell(h, a_U) + R_U^\ell(h^*, a_U) + R_P^\ell(h^*, a_P) \\
&= D_{h,\mathcal{H}}^f(P_U \| P_P) + R_U^\ell(h) + R_U^\ell(h^*) + R_P^\ell(h^*)
\end{aligned}
$$

### E.4 PROOF OF THEOREM 4.6 AND THE BENEFIT OF OUR DESIGN

Combining Theorem 4.5, Lemma 4.4 and the property of Rademacher Complexity, we can easily get:

$$
\begin{aligned}
R_P^l(h) - R_U^l(h) &\leq D_{h,\mathcal{H}}^f(U \| P) \\
&\quad + \hat{R}_U^l(h^*) + 4\mathfrak{R}_U(\ell \circ \mathcal{H}) + 2\sqrt{\frac{\log \frac{1}{\delta}}{2m}} \\
&\quad + \hat{R}_P^l(h^*) + 2(L+1)\mathfrak{R}_P(\ell \circ \mathcal{H}) + 2\sqrt{\frac{\log \frac{1}{\delta}}{2n}}
\end{aligned}
$$

Since by definition we have $D_{h,\mathcal{H}}^f(U \| P) \leq D_f(U \| P)$, and for $D_f(U \| P) = \text{TV}(U \| P)$, we have:

$$
\begin{aligned}
R_P^l(h) - R_U^l(h) &\leq \text{TV}(U \| P) \\
&\quad + \hat{R}_U^l(h^*) + 4\mathfrak{R}_U(\ell \circ \mathcal{H}) + 2\sqrt{\frac{\log \frac{1}{\delta}}{2m}} \\
&\quad + \hat{R}_P^l(h^*) + 2(L+1)\mathfrak{R}_P(\ell \circ \mathcal{H}) + 2\sqrt{\frac{\log \frac{1}{\delta}}{2n}}.
\end{aligned}
\tag{13}
$$

Now we motivate why minimizing the objective $\mathcal{L}_{\text{FAC}}$ leads to small $\text{TV}(U \| P)$. Let $U, P$ be the empirical distributions over the common measurable space $\mathcal{X} := \{z_j^U\}_{j=1}^n \cup \{z_k^P\}_{k=1}^m$ with densities $\hat{p}_U, \hat{p}_P$ that are $c_U, c_P$-Lipschitz with respect to $\ell_2$-norm, respectively. Let $x^* := \arg\min_{x \in \mathcal{X}} |\hat{p}_U(x) - \hat{p}_P(x)|$, $\delta := |\hat{p}_U(x^*) - \hat{p}_P(x^*)|$, and

$$
\sigma := \sum_{x \in \mathcal{X}} \|x - x^*\| = \sum_{x \in \mathcal{X}} \sqrt{2 - 2\log \text{sim}(x, x^*)},
$$

where the equality is due to law of cosine (and that $\text{sim}$ normalizes $z_j$). We first show how $\text{TV}(U \| P)$ is related to $\delta$ and $\sigma$.

**Lemma E.1.**

$$
\text{TV}(U \| P) \leq \frac{1}{2}\left(|\mathcal{X}|\delta + (c_U + c_P)\sigma\right).
$$

*Proof.*

$$TV(U\|P) \coloneqq \frac{1}{2} \sum_{x \in \mathcal{X}} |\hat{p}_U(x) - \hat{p}_P(x)|$$

$$\leq \frac{1}{2} \sum_{x \in \mathcal{X}} |\hat{p}_U(x) - \hat{p}_U(x^*)| + |\hat{p}_U(x^*) - \hat{p}_P(x^*)| + |\hat{p}_P(x^*) - \hat{p}_P(x)| \quad \text{(triangle inequality)}$$

$$= \frac{1}{2} \left( |\mathcal{X}|\delta + \sum_{x \in \mathcal{X}} |\hat{p}_U(x) - \hat{p}_U(x^*)| + |\hat{p}_P(x) - \hat{p}_P(x^*)| \right)$$

$$\leq \frac{1}{2} \left( |\mathcal{X}|\delta + (c_U + c_P) \sum_{x \in \mathcal{X}} \|x - x^*\| \right) \quad \text{(Lipschitz conditions)}$$

$$= \frac{1}{2} \left( |\mathcal{X}|\delta + (c_U + c_P)\sigma \right).$$

$\square$

Next we motivate why minimizing our objective $\mathcal{L}_{\text{FAC}}$ leads to small $\delta$ and $\sigma$ simultaneously, hence small $TV(U\|P)$. Recall that our fairness-aware contrastive loss is

$$\mathcal{L}_{\text{FAC}} \coloneqq \mathcal{L}_{\text{fair}} + \mathcal{L}_{\text{unif}},$$

where

$$\mathcal{L}_{\text{fair}} \coloneqq -\log \left( \sum_{j \in [n]} \sum_{k \in [m]} \text{sim} \left( z_j^U, z_k^P \right) \right),$$

$$\mathcal{L}_{\text{unif}} \coloneqq \log \left( \sum_{j \neq k} \text{sim} \left( z_j^U, z_k^U \right) + \sum_{j \neq k} \text{sim} \left( z_j^P, z_k^P \right) \right).$$

Intuitively, minimizing $\mathcal{L}_{\text{FAC}}$ leads to small $\mathcal{L}_{\text{fair}}$ and $\mathcal{L}_{\text{unif}}$ simultaneously, which correspond to large $\text{sim}(z_j^U, z_k^P)$ and small $\text{sim}(z_j^U, z_k^U), \text{sim}(z_j^P, z_k^P)$, which in turn correspond to small $\|z_j^U - z_k^P\|$ and large $\|z_j^U - z_k^U\|, \|z_j^P - z_k^P\|$. Hence it is natural to consider the following surrogate losses

$$\mathcal{L}'_{\text{fair}} \coloneqq \sum_{j,k \in [n]} \|z_j^U - z_k^P\|,$$

$$\mathcal{L}'_{\text{unif}} \coloneqq -(\sum_{j \neq k} \|z_j^U - z_k^U\| + \|z_j^P - z_k^P\|).$$

Then it follows immediately that $\sigma \leq \mathcal{L}'_{\text{fair}}$, explaining why minimizing our objective $\mathcal{L}_{\text{FAC}}$ (hence $\mathcal{L}'_{\text{fair}}$) leads to small $\sigma$.

To see that $\delta \coloneqq |\hat{p}_U(x^*) - \hat{p}_P(x^*)|$ cannot be too large, first consider the extreme case where $\{z_j^U\}_{j=1}^n \cap \{z_k^P\}_{k=1}^n = \emptyset$. Without loss of generality let $\|z_1^U - z_1^P\| = \max_{j,k \in [n]} \|z_j^U - z_k^P\|$. Then adjusting $z_1^U, z_1^P$ to be the unit vector on their angle bisector clearly decreases $\mathcal{L}'_{\text{fair}}$ without affecting $\mathcal{L}'_{\text{unif}}$ by much due to high uniformity within $\{z_j^U\}_{j=1}^n$ and $\{z_k^P\}_{k=1}^n$ respectively. Hence we may assume without loss of generality that $z_1^U = z_1^P = x^*$. Next consider the extreme case where $\hat{p}_U(x^*) = \frac{1}{n}$ and $\hat{p}_P(x^*) = 1$. Then adjusting $z_2^P = \arg\max_{x \neq x^*} \sum_{j \in [n]} \|x - z_j^U\|$ clearly decreases $\mathcal{L}'_{\text{unif}}$ without affecting $\mathcal{L}'_{\text{fair}}$ by much due to high uniformity within $\{z_j^U\}_{j=1}^n$. Hence minimizing our objective $\mathcal{L}_{\text{FAC}}$ leads to small $\delta \coloneqq |\hat{p}_U(x^*) - \hat{p}_P(x^*)|$.

## F ADDITIONAL EXPERIMENTS

### F.1 TRAINING DETAILS AND EXPERIMENTAL SETUP

For the COMPAS dataset, we use a two-layer MLP with hidden units of [32, 32]. For all the other datasets, we use MLP with one hidden layer of dimension 128. For FADIG, we set the hyperparameter $\alpha = 4$ across all the data sets and use the Adam optimizer. For the baselines, we use the

suggested hyperparameter settings in their original papers. For the four independent runs, we choose random seeds in [40, 41, 42, 3407]. All our experiments were executed using one Tesla V100 SXM2 GPUs, supported by a 12-core CPU operating at 2.2GHz.

For evaluation, since the task is unsupervised, the train and test sets are the same. Following Shekhar et al. (2021); Zhang & Davidson (2021); Ahmed et al. (2021), to evaluate task performance, we use Recall@K and ROCAUC. For fairness evaluation, considering the imbalance between normal examples and anomalies, we focus on Recall Parity in anomaly detection. Given our score-based anomaly detection framework, we would like to state the mathematical formulation of Recall Parity fairness in anomaly detection as: Let anomaly score for example $x$ be $s(x)$ and let $t_K$ be the anomaly score threshold for top-K selection. Then, the predicted normal examples are the ones with $s(x) < t_K$, and the predicted anomalies are those with $s(x) \geq t_K$. The recall parity in anomaly detection requires that $P(s(x) \geq t_K | x \in U, y = 1) = P(s(x) \geq t_K | x \in P, y = 1)$. We use the absolute value of their difference, *i.e.*, Recall Diff, to evaluate the fairness level.

## F.2 MORE EFFECTIVENESS VALIDATION OF FADIG UNDER DIFFERENT K

We also conduct experiments on the four datasets with different choices of $K$, and the results are in Table 10 and Table 11. The AUCROC scores are the same as in the main paper. We can also tell from the tables that accuracy difference is inadequate for measuring group fairness in the imbalanced setting.

Table 10: Performance on Image Datasets.

| Methods | MNIST-USPS (K=1000) | | | MNIST-Invert (K=400) | | |
|---|---|---|---|---|---|---|
| | Recall@K | Acc Diff | Rec Diff | Recall@K | Acc Diff | Rec Diff |
| FairOD | 10.46±1.16 | 4.35±0.33 | 13.21±1.43 | 6.05±0.21 | 2.70±0.15 | 9.99±1.18 |
| DCFOD | 10.24±0.82 | 4.79±1.12 | 8.40±1.83 | 5.57±1.70 | 2.69±0.37 | 8.78±2.31 |
| FairSVDD | 13.75±1.83 | 5.73±5.64 | 13.49±2.55 | 10.57±0.92 | 5.38±3.12 | 14.25±2.96 |
| MCM | 34.38±0.32 | 29.81±0.84 | 52.46±0.94 | 22.48±0.54 | 8.32±1.10 | 64.37±1.66 |
| NSNMF | 33.56±0.70 | 22.26±0.40 | 65.12±2.36 | 43.91±0.84 | 4.54±0.20 | 55.20±0.92 |
| Recontrast | 45.73±2.74 | 10.59±2.62 | 29.62±2.40 | 52.00±4.86 | 13.81±4.30 | 54.96±13.77 |
| FADIG | 61.60±2.50 | 6.50±0.89 | 7.95±5.94 | 62.28±3.24 | 1.62±1.32 | 7.02±4.48 |

Table 11: Performance on Tabular Datasets

| Methods | COMPAS (K=300) | | | CelebA (K=4500) | | |
|---|---|---|---|---|---|---|
| | Recall@K | Acc Diff | Rec Diff | Recall@K | Acc Diff | Rec Diff |
| FairOD | 14.20±1.83 | 3.92±1.63 | 10.75±0.90 | 7.95±0.21 | 4.94±0.25 | 2.26±1.06 |
| DCFOD | 13.10±1.35 | 3.57±2.29 | 7.23±2.82 | 8.64±0.79 | 4.98±0.40 | 9.24±1.12 |
| FairSVDD | 13.02±1.66 | 3.90±2.43 | 9.45±3.80 | 8.82±0.61 | 2.21±0.40 | 10.22±2.33 |
| MCM | 16.87±1.14 | 4.10±1.98 | 10.17±1.64 | 9.26±0.48 | 7.21±5.98 | 28.69±12.14 |
| NSNMF | 17.29±1.42 | 3.60±1.93 | 33.57±1.22 | 8.90±1.09 | 5.66±0.54 | 40.51±1.54 |
| FADIG | 19.14±2.29 | 9.35±3.00 | 4.75±3.69 | 10.56±1.11 | 13.04±0.30 | 5.10±1.52 |

## F.3 COMPARISON WITH REWEIGHTING HEURISTIC

We compare our design of automatic re-balancing with a simple heuristic of setting $\epsilon$ as the scaled majority group size, and the results on the four datasets are shown in Table 12. We can observe that compared with the heuristic weight, our designed learnable $\epsilon$ can effectively better enhance task performance and meanwhile promote fairness.

Table 12: Comparison of our designed rebalancing strategy with group ratio weighting.

| Datasets | FADIG | | | | Group Ratio | | | |
|---|---|---|---|---|---|---|---|---|
| | Recall@K | ROCAUC | Rec Diff | Time(s) | Recall@K | ROCAUC | Rec Diff | Time(s) |
| MNIST-USPS | 67.16±0.37 | 91.27±0.49 | 3.73±2.13 | 122.84 | 62.35±0.10 | 87.61±0.47 | 11.87±3.90 | 75.28 |
| MNIST-Invert | 72.37±0.32 | 98.03±0.01 | 6.75±0.34 | 52.28 | 68.33±0.24 | 89.91±0.02 | 74.22±0.26 | 146.45 |
| COMPAS | 34.43±0.42 | 61.85±0.52 | 5.81±4.36 | 17.94 | 33.42±1.61 | 60.49±4.20 | 5.85±5.75 | 15.95 |
| CelebA | 11.94±0.67 | 59.41±0.58 | 4.66±1.72 | 52.81 | 12.75±0.62 | 57.23±0.25 | 13.54±0.89 | 48.12 |

## F.4 PARAMETER ANALYSIS

In this section, we conduct the parameter analysis on the four datasets. The experiments are repeated four times and the mean of the recall rate and recall difference are reported. Figure 5 shows the parameter analysis for the parameter $\alpha$ on the four datasets, respectively. The parameter $\alpha$ is used to balance the importance between the reconstruction error and the fair contrastive loss. We can observe that our method is robust to the choice of $\alpha$, which may be a benefit from our designed re-balancing autoencoder.

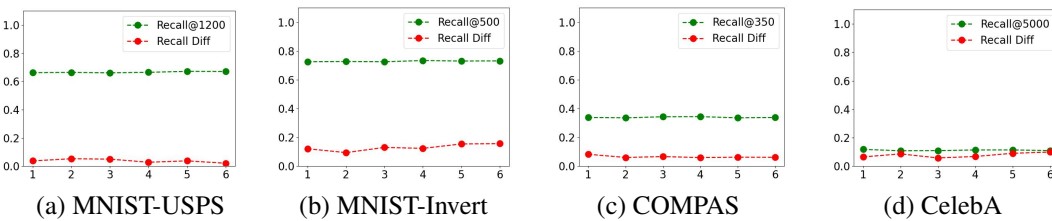

(a) MNIST-USPS     (b) MNIST-Invert     (c) COMPAS     (d) CelebA

Figure 5: Parameter Analysis of $\alpha$ on four datasets. The x-axis is $\alpha$ and the y-axis is for the values of recall and recall difference.

## F.5 DIFFERENT ANOMALY TYPES

We extend our experimental setup to analyze how our method performs on different types of anomalies. In MNIST-USPS and MNIST-Invert, the normal samples are digit 0 and the anomalies are the digits 1-9. In COMPAS and CelebA, we use whether the sample is reoffending / attractive or not to define normal and abnormal samples. Compared with the two image datasets, the anomalies in the tabular ones are more clustered. Thus, we sample more clustered anomalies on the image data set MNIST-USPS by selecting only digit 1 as the anomalies with the same anomaly amount. The results are shown in Table 13. We can observe that our proposed method achieves the best recall rate and the second-best ROCAUC score, with a relatively low recall difference. Notably, the baselines with extremely low recall differences are showing "fake" fairness since their task performances are very poor.

Table 13: Performance on more clustered anomaly detection.

| Methods | Recall@K | ROCAUC | Rec Diff | Time(s) |
|---|---|---|---|---|
| FairOD | 12.03±0.42 | 50.04±0.33 | 3.99±3.89 | 39.06 |
| DCFOD | 12.40±1.23 | 49.94±0.72 | 4.08±6.02 | 757.88 |
| FairSVDD | 18.67±1.73 | 54.05±9.09 | 29.72±20.39 | 212.38 |
| MCM | 4.01±0.46 | 14.28±0.60 | 3.81±1.57 | 51.28 |
| NSNMF | 4.64±0.11 | 45.92±0.78 | 12.46±0.62 | 71.61 |
| Recontrast | 16.34±1.78 | 51.82±2.26 | 40.81±18.42 | 259.77 |
| FADIG | 21.04±1.27 | 53.91±1.22 | 14.39±0.43 | 415.17 |

F.6 ADDITIONAL COMPARISON WITH DATA IMBALANCE BASELINE

We further compare our method with a classical method handling data imbalance for tabular data, SMOTE (Chawla et al., 2002). The results on the COMPAS and CelebA data sets are shown in Table 14. We can see that FADIG outperforms it in both task performance and fairness level.

Table 14: Additional results on tabular datasets. The best score is marked in bold.

| Methods | COMPAS (K=350) | | | | CelebA (K=5000) | | | |
|---|---|---|---|---|---|---|---|---|
| | Recall@K | ROCAUC | Rec Diff | Time(s) | Recall@K | ROCAUC | Rec Diff | Time(s) |
| FairOD | 16.56±2.12 | 50.09±1.28 | 7.97±1.23 | 4.18 | 8.93±0.14 | 49.94±0.12 | **0.68±0.56** | 78.92 |
| DCFOD | 16.08±1.94 | 49.55±1.21 | 9.81±1.76 | 115.86 | 9.66±0.69 | 49.92±0.14 | 7.83±1.26 | 2517.68 |
| FairSVDD | 15.33±2.10 | 52.68±5.29 | 11.57±4.06 | 6.81 | 10.19±0.50 | 58.40±1.02 | 10.95±1.93 | 243.17 |
| MCM | 21.10±0.54 | 50.97±0.43 | 6.29±2.66 | 38.12 | 11.03±0.38 | 46.23±3.46 | 26.15±9.31 | 640.12 |
| NSNMF | 22.92±0.32 | 57.97±0.66 | 36.78±1.71 | 7.69 | 10.91±0.54 | 50.45±0.30 | 8.04±1.33 | 1927.55 |
| SMOTE | 29.92±2.62 | 60.45±4.18 | 6.96±5.75 | 8.91 | 8.14±0.40 | 45.39±0.35 | 5.07±1.43 | 332.17 |
| FADIG | **34.38±0.36** | **61.45±0.47** | **5.97±4.34** | 19.88 | **11.96±0.49** | **59.43±0.42** | 4.72±1.26 | 48.93 |

F.7 GRAPH TASKS

We also compare FADIG on graph tasks with two graph anomaly detection baselines, DOMI-NANT (Ding et al., 2019) and GRADATE (Duan et al., 2023). We adapt our method on the graph dataset Flickr (Li et al., 2015), replacing the backbone with GCN. The results are shown in Table 15. We can observe that FADIG outperforms DOMINANT in both task performance and fairness. While GRADATE has better task performance compared with our method, it may be because we have not optimized our framework specifically for graph data. In addition, our method achieves a much lower recall difference than both of the baselines.

Table 15: Performance on the graph dataset.

| Methods | Recall@K | ROCAUC | Rec Diff |
|---|---|---|---|
| DOMINANT | 21.34±0.48 | 61.72±0.59 | 20.56±3.32 |
| GRADATE | 24.96±0.62 | 66.54±1.12 | 35.63±5.34 |
| FADIG | 23.10±0.61 | 63.89±1.12 | 5.33±1.52 |

# G LIMITATIONS AND BROADER IMPACT

This paper proposes a fairness-aware anomaly detection method, which aims to provide fair results when the algorithm is applied to detect anomalies. Our method currently focus on the binary group fairness case. We can naturally extend our framework to the multi-attribute case by encouraging the similarity among the groups. Incoporating individual fairness notions would be an interesting future direction. By embedding fairness into anomaly detection algorithms, this work contributes to reducing bias and discrimination in AI applications, ensuring that technologies serve diverse populations equitably. In sectors such as finance, healthcare, and law enforcement, where anomaly detection plays a crucial role in identifying fraud, diseases, and criminal activities, incorporating fairness principles can prevent the perpetuation of historical biases and protect vulnerable groups from unjust outcomes. Furthermore, by advancing fairness in AI, this research aligns with global efforts to promote ethics in technology development, fostering trust between AI systems and their users.

