# OpenReview forum: "Fair Anomaly Detection For Imbalanced Groups"
_ICLR.cc/2025/Conference — Submitted to ICLR 2025_

### Official Review · Reviewer_JisA · 2024-10-23

**Soundness:** 3
**Presentation:** 3
**Contribution:** 3
**Rating:** 6
**Confidence:** 2

**Summary:**

This paper develops a fair anomaly detection model based on reconstruction-based detection. A fairness-aware contrastive learning and a re-balancing autoencoder are developed to ensure fairness.

**Strengths:**

- This paper targets an important but still under-exploited problem: fair anomaly detection.

- The proposed approach can achieve a better trade-off between utility and fairness.

**Weaknesses:**

- Maybe I am missing something. Section 4 looks incomplete to me. It demonstrates the fairness with Rademacher Complexity, but I am not sure how the proposed approach can minimize the upper bound given in Theorem 4.6. That said, it would be better to provide a more explicit explanation of how the proposed components (the fairness-aware contrastive learning or re-balancing autoencoder) relate to minimizing the terms in the upper bound given in Theorem 4.6.

- Theorem 4.5 states that the *risk difference* between the two groups is upper bounded by Eq 10. Are there any connections between the risk difference in terms of Rademacher complexity and risk difference in terms of fairness metrics, such as the recall-based metrics used in experiments?

    - Please clarify whether there's a theoretical relationship between the risk difference bound and the recall difference metric used in the experiments.


- Overall, Section 4 looks more like a general theoretical analysis. It would be better to explicitly discuss how to leverage the theorems in Sec 4 to analyze the fairness issue in anomaly detection tasks, especially the reconstruction-based anomaly detection models.

**Questions:**

Please check my comments above.

---

> ### Author Response · Authors · 2024-11-24
> **Response to Reviewer JisA**
>
> Thank you very much for your detailed and constructive comments. We are glad to provide the following responses to your questions.
>
> `Weakness 1: Unclear how the proposed approach can minimize the upper bound given in Theorem 4.6. `
>
> Answer 1: In Appendix E.4, we present the detailed proof of how our designed $\mathcal{L}_\text{FAC}$ minimizes the upper bound in Theorem 4.6.
>
> `Weakness 2 & 3: Please clarify whether there is a theoretical relationship between the risk difference bound and the recall difference metric used in the experiments. It would be better to explicitly discuss how to leverage the theorems in Sec 4 to analyze the fairness issue in anomaly detection tasks, especially the reconstruction-based anomaly detection models.`
>
> Answer 2 & 3 : The risk difference bound places a theoretical upper limit on disparities in recall, assuming the loss function explicitly penalizes false negatives. If risk differences are bounded tightly, recall differences are also constrained, since both rely on group-level error dynamics. While recall difference focuses specifically on positive cases, the risk difference bound incorporates all types of errors (false positives, false negatives, etc.). Therefore, a high risk difference often correlates with high recall disparities, but the risk metric provides a broader error perspective. And that is why we provide a risk difference bound for the fairness guarantee of our fair anomaly detection method.

---

### Official Review · Reviewer_oKCG · 2024-11-01

**Soundness:** 2
**Presentation:** 2
**Contribution:** 2
**Rating:** 3
**Confidence:** 3

**Summary:**

The authors propose a new fairness-aware anomaly detection method called FADIG. Their method uses contrastive learning and a modified, reweighted, autoencoder in order to detect anomalies in datasets with multiple groups of differing representation.

**Strengths:**

The paper clearly states its purpose. The mathematics of the paper are fairly easy to follow and clearly substantiate the intuitive reasoning of the method. The proposal of the authors should also be lauded for its relative simplicity which achieves (seeminly) good results.

**Weaknesses:**

The points below are given in no particular ordering of importance (within category):

**Major:**
- The authors state in the second paragraph of the introduction that deep anomaly detection methods demonstrate significantly better performance than shallow anomaly detection methods. While this can easily be argued for certain applications such as computer vision, it is certainly not true for tabular data, which is also analyzed within this paper and several of the other methods benchmarked in this study. See for example recent comparisons in ADBench by Han et al. (2022) and Bouman et al. (2024) in "Unsupervised Anomaly Detection Algorithms on Real-world Data: How Many Do We Need?".
- The use of an autoencoder as an anomaly detector relies heavily on an assumption stated by the authors that " Anomalies tend to exhibit large reconstruction errors because they do not conform to the patterns in the data as coded by the autoencoder." This is an assumption which has been heavily questioned in recent studies, such as:
	- Nicholas Merrill and Azim Eskandarian. Modified autoencoder training and scoring for robust unsupervised anomaly detection in deep learning. IEEE Access, 8:101824–101833, 2020.
	- Laura Beggel, Michael Pfeiffer, and Bernd Bischl. Robust anomaly detection in images using adversarial autoencoders. In Machine Learning and Knowledge Discovery in Databases: European Conference, ECML PKDD 2019, W¨urzburg, Germany, September 16–20, 2019, Proceedings, Part I, pp. 206–222. Springer, 2020
	- Marcella Astrid, Muhammad Zaigham Zaheer, Jae-Yeong Lee, and Seung-Ik Lee. Learning not to reconstruct anomalies. arXiv preprint arXiv:2110.09742, 2021.
	- Marcella Astrid, Muhammad Zaigham Zaheer, Djamila Aouada, and Seung-Ik Lee. Exploiting autoencoder’s weakness to generate pseudo anomalies. Neural Computing and Applications, pp. 1–17, 2024
	- Dong Gong, Lingqiao Liu, Vuong Le, Budhaditya Saha, Moussa Reda Mansour, Svetha Venkatesh, and Anton van den Hengel. Memorizing normality to detect anomaly: Memory-augmented deep autoencoder for unsupervised anomaly detection. In Proceedings of the IEEE/CVF international conference on computer vision, pp. 1705–1714, 2019.
	- Bo Zong, Qi Song, Martin Renqiang Min, Wei Cheng, Cristian Lumezanu, Daeki Cho, and Haifeng Chen. Deep autoencoding gaussian mixture model for unsupervised anomaly detection. In International conference on learning representations, 2018
- The authors assume in section 3.2 that the model is only capable of fitting 2 out of 4 subgroups. Yet this seems like a strong assumption with little theoretical foundation.
- For the CelebA dataset, all detectors perform much worse than should be expected, with almost all detectors being close to random. This indicates that perhaps it is not a well-suited benchmark dataset, or other detectors should be shown which do perform well.



**Minor:**
- The connection to relevant literature is too sparse. Applications and other studies into fairness of anomaly detectors should be elaborated upon in more detail during the introduction.
- The captions of figures are somewhat shorthand. I would advise the authors to rethink the captions so they allow for the figures to be understood more easily as-as, rather than needing the full context of the section or paper.
- The introduction currently contains a preview of the results, which disturbs the flow of the paper. I'd advise to move this to the result section in full.

**Questions:**

- I'm missing a connection of the fairness discussion, and the presence of multiple groups, to the more classical literature on anomaly detection. Specifically anomaly detection in the presence of multiple groups has been studied extensively, and relates to the concept of "local" and "global" anomalies, see for example "LOF: identifying density-based local outliers" by Breunig et al., "LoOP: local outlier probabilities" by Kriegel et al., other work by Kriegel et al. throughout the 00's and 10' as well as a recent examination by Bouman et al. in "Unsupervised Anomaly Detection Algorithms on Real-world Data: How Many Do We Need?". Can the authors elaborate on this connection?
- Figure 2 is currently hard to understand. What representation is being shown? Why is the data circularly constrained, which dataset is shown, what do the x and y-axis represent? Consider revising this.
- What does the FADIG abbreviation mean? The beginning of section 3 implies an abbreviation due to the capitalization, but it does not match up with the letters and is therefore slightly confusing.
- It currently still eludes me how the model is applied in practice in the unsupervised setting. Specifically, without knowledge of whether a sample belongs to the protected or unprotected group at train time, how is the loss calculated? It seems that we need access to the protected and unprotected loss, but within the unsupervised setting we would never be able to differentiate between the two. Could the authors elaborate further?
- In the characteristics of the datasets, it seems that the number of anomalies for the 4 datasets is 1205, 479, 364, and 5150. Yet, the k-values chosen for the results presented are not at these values, but rather at values close. Where does this difference come from? I would like to remark that this is not in line with other measures such as the r-precision, which are calculated with k being the exact number of anomalies in the dataset.
- Some detectors do not perform well on any dataset. Could the authors elaborate on how this could happen? I've noted here that for example FairOD uses fully different benchmarking datasets, why not include these for this study?
- The chosen architecture for the used autoencoder is fairly shallow, and notable barely compresses the data. How were these architectures chosen, and how did the authors ensure that they did not overfit on the proposed benchmark? Similarly, could more details on other hyperparameters and how they were chosen be provided?
- In the reproducibility section, the authors list the random seeds that were used. These do not seem structured in either a logical, or a random way, which leads me to suspect that some, albeit accidental, cherrypicking may have occured. Could the authors elaborate on this?

---

> ### Author Response · Authors · 2024-11-24
> **Response to Major Weakness**
>
> Thank you very much for your detailed and constructive comments. We are glad to provide the following responses to your questions.
>
> `Major Weakness 1: Question in the statement 'deep anomaly detection methods demonstrate significantly better performance than shallow anomaly detection methods.'`
>
> Answer 1: We agree with you that shallow models might have better performance on some easy datasets, such as tabular data. We mean to say that deep models tend to outperform shallow model in some complex datasets. We have revised our introduction by emphasizing that usually deep anomaly detection methods perform better than the shallow methods on more complicated tasks.
>
> `Major Weakness 2 & 3: The use of an autoencoder as an anomaly detector relies heavily on an assumption stated by the authors that " Anomalies tend to exhibit large reconstruction errors because they do not conform to the patterns in the data as coded by the autoencoder." This is an assumption which has been heavily questioned in recent studies. The authors assume in section 3.2 that the model is only capable of fitting 2 out of 4 subgroups. Yet this seems like a strong assumption with little theoretical foundation.`
>
> Answer 2&3: There are several reasons for us to choose vanilla autoencoder as the anomaly detector.
> 1. Vanilla autoencoder is simple yet effective to identify the anomaly from both protected and unprotected group. In the early stage of the algorithm development, we try to incorporate the idea of the MemAE [1] by using it as the backbone for our method. However, the experimental results show that the performance of MemAE tends to have worse performance than a vanilla autoencoder architecture on most datasets.
> 2. We try to avoid the over-fitting issue by using vanilla autoencoder, specifically for the tabular data with less than 10 features.
> 3. The theoretical analysis is built upon the assumption that the model is only capable of capturing the normal samples for protected and unprotected group, and thus the anomalies from both groups will have the larger reconstruction error. Otherwise, the model cannot distinguish anomalies from normal samples. However, a complex model could not fulfill such an assumption. Thus, we select vanilla autoencoder as the backbone for our method.
> 4. We consider the efficiency of the vanilla autoencoder over the complex backbone in other anomaly detection methods.
> 5. We would like to point out that "Anomalies tend to exhibit large reconstruction errors " is still a widely accepted assumption in the field of anomaly detection, such as [2][3][4].
>
> [1] Dong Gong, Lingqiao Liu, Vuong Le, Budhaditya Saha, Moussa Reda Mansour, Svetha Venkatesh, and Anton van den Hengel. Memorizing normality to detect anomaly: Memory-augmented deep autoencoder for unsupervised anomaly detection. In Proceedings of the IEEE/CVF international conference on computer vision, pp. 1705–1714, 2019.
>
> [2] Guo, Jia, Lize Jia, Weihang Zhang, and Huiqi Li. "Recontrast: Domain-specific anomaly detection via contrastive reconstruction." Advances in Neural Information Processing Systems 36 (2023).
>
> [3] Z. You et al., “A unified model for multi-class anomaly detection,” in Advances in Neural Information Processing Systems (S. Koyejo, S. Mohamed, A. Agarwal, D. Belgrave, K. Cho, and A. Oh, eds.), vol. 35, pp. 4571–4584, Curran Associates, Inc., 2022.
>
> [4] H. Deng and X. Li, “Anomaly Detection via Reverse Distillation from One-Class Embedding,” in Proceedings of the IEEE/CVF Conference on Computer Vision and Pattern Recognition, pp. 9737–9746, 2022.
>
> `Major Weakness 4: For the CelebA dataset, all detectors perform much worse than should be expected, with almost all detectors being close to random. This indicates that perhaps it is not a well-suited benchmark dataset, or other detectors should be shown which do perform well.`
>
>
> Answer 4: We agree that CelebA dataset is a very challenging dataset.
> Following [5], we evaluate the performance of all detectors on this dataset. Notice that [5] assumes that the label information is partially available for samples in the training set and [5] achieves 95.1% AUC-ROC on this dataset. While all detectors do not perform well on this dataset in the unsupervised setting, it is still important to evaluate the performance of anomaly detection methods on this benchmark dataset in the field of fair machine learning.
>
> [5] Pang, Guansong, Chunhua Shen, and Anton Van Den Hengel. "Deep anomaly detection with deviation networks." In Proceedings of the 25th ACM SIGKDD international conference on knowledge discovery \& data mining, pp. 353-362. 2019.

---

> > ### Author Response · Authors · 2024-11-24
> > **Response to Minor Weaknesses and Questions**
> >
> > `Minor Weakness 1&2: Suggestions in relevant literature and figure caption.`
> >
> > Answer 1&2: Thanks for your suggestion. We have revised the figure caption and the literature review.
> >
> > `Minor Weakness 3: The introduction currently contains a preview of the results, which disturbs the flow of the paper. I'd advise moving this to the result section in full.`
> >
> > Answer 3: We appreciate your suggestion. However, we would like to point out that using the preview of the results in the introduction serves as a motivation for us to rethink about the issue of the current fair anomaly detection methods. In addition, including experimental results in the introduction as the motivation is pretty common in many famous and well-cited works, such as [1] [2].
> >
> > [1] Khosla, Prannay, Piotr Teterwak, Chen Wang, Aaron Sarna, Yonglong Tian, Phillip Isola, Aaron Maschinot, Ce Liu, and Dilip Krishnan. "Supervised contrastive learning." Advances in neural information processing systems 33 (2020): 18661-18673.
> >
> > [2] Chen, Ting, Simon Kornblith, Mohammad Norouzi, and Geoffrey Hinton. "A simple framework for contrastive learning of visual representations." In International conference on machine learning, pp. 1597-1607. PMLR, 2020.
> >
> >
> > `Question 1: The connection of the fairness discussion, and the presence of multiple groups, to the more classical literature on anomaly detection. `
> >
> > Answer 1: We appreciate your suggestion. We have added these works in our discussion of related work in Appendix B.
> >
> > `Question 2: Figure 2 is currently hard to understand. `
> >
> > Answer 2: Sorry for the confusion. We have revised Figure 2 and the caption. Figure 2 is an illustration of different data representations in a unit sphere. In Figure 2, we try to show that only representations which consider both fairness and uniformity can succeed in the fair anomaly detection task.
> >
> > `Question 3: What does the FADIG abbreviation mean?`
> >
> > Answer 3: Thanks for pointing this out. We have underlined each letter from the name in the introduction section.
> >
> > `Question 4: In the unsupervised setting, how can we access the protected and unprotected group information?`
> >
> > Answer 4: We would like to clarify that **the label information is different from the group information**. In the unsupervised setting, we do not have access to the label information (i.e., the example is normal or anomaly), but we do have the group information of the training data. Besides, during the test time, we do not need the group information.
> >
> > `Question 5: How are different ks for different datasets chosen?`
> >
> > Answer 5: We would like to point out that in the unsupervised setting, the exact number of anomalies is unknown and we have to approximate a number that is close to the exact number of the anomalies. We also test different $k$ in Appendix F.3, and we can observe that with different choices of $k$, our method still outperforms the baselines in both task performance and fairness.
> >
> > `Question 6: Question about the baseline performance and choice of datasets.`
> >
> > Answer 6: We would like to point out that FairOD [3] was only tested on tabular datasets. To evaluate the performance of our method on various types of data, we test our method on both tabular and image datasets following (Zhang & Davidson, 2021). Besides, we test on different anomaly types in Appendix F.5, and further compare our method on graph datasets in Appendix F.7.
> >
> >
> > `Question 7: Question about how to choose the architecture for the used autoencoder.`
> >
> > Answer 7: We would like to point out that we follow the architecture of FairOD [3], which uses two-layer MLP to learn the representation. Another reason of using shallow architecture is to avoid overfitting, specifically for limited number of features on some datasets, e.g., 6 features on the COMPAS dataset and 39 features on the CelebA dataset.
> >
> > [3] Shubhranshu Shekhar, Neil Shah, and Leman Akoglu. Fairod: Fairness-aware outlier detection. In Proceedings of the 2021 AAAI/ACM Conference on AI, Ethics, and Society, pp. 210–220, 2021
> >
> > `Question 8: Question about the random seeds in the reproducibility section.`
> >
> > Answer 8: The random seeds are chosen randomly given the author's coding style (some people like using 1024, 1111, etc.). We do not observe any consistent performance increase/decrease under certain random seeds, either in our method or in the baselines.

---

> > > ### Comment · Reviewer_oKCG · 2024-11-25
> > >
> > > Thanks again for the elaboration.
> > > Like I mentioned for the major weaknesses, I would like to see these elaborations included in the manuscript.
> > >
> > > I will reply to each of your responses below where applicable:
> > >
> > > **Minor Weaknesses**
> > >
> > > - 3. I respect the choice of the authors to include some results in the introduction.
> > >
> > > **Questions**
> > > - 5. It remains unclear to me how the exact k's were chosen in this specific study. In many cases they are close,  but the selection strategy is unclear to me, and thus does not provide guidelines to potential users.
> > >
> > > - 6. It seems I should further elaborate: FairOD specifically uses **more** benchmark datasets for the tabular setting. Why where these not included in this study as well?
> > >
> > > - 7. It seems to me that the biggest issue in overfitting is a bottleneck which is too large, not an increasing number of layers. So then the choice of architecture still needs to be motivated further.
> > >
> > > - 8. This does not sufficiently alleviate my concerns, but I want to thank the authors for elaborating nonetheless.

---

> > > > ### Author Response · Authors · 2024-12-04
> > > > **Response to follow-up questions**
> > > >
> > > > `Q5: It remains unclear to me how the exact k's were chosen in this specific study. In many cases they are close, but the selection strategy is unclear to me, and thus does not provide guidelines to potential users.`
> > > >
> > > > Answer: For the dataset we use, we approximate k as a number that is close to the exact number of the anomalies in hundreds or fifties. For potential users, with knowledge of the dataset (i.e., exact numbers of the anomalies), you can choose the number in hundreds closest to the the exact number of anomalies. If you do not know the exact number, with some domain knowledge, you can set k according to the approximated proportion of anomalies in the dataset. E.g., you can set k as 10% of the total sample number if you expect such an anomaly rate in this dataset.
> > > >
> > > > `Q6: FairOD specifically uses more benchmark datasets for the tabular setting. Why were these not included in this study as well?`
> > > >
> > > > Answer: Thank you for your suggestion. We are happy to add comparisons on more tabular benchmark datasets in the appendix.
> > > >
> > > > `Q7: It seems to me that the biggest issue in overfitting is a bottleneck which is too large, not an increasing number of layers. So then the choice of architecture still needs to be motivated further.`
> > > >
> > > > Answer: We use the common choice of the mlp design on the datasets. We would like to clarify that we are not optimizing the base architecture for anomaly detection, thus the design of the architecture is out of the scope of this work. We would like to add a case study of the results with different architecture choices in the appendix.
> > > >
> > > > `Q8: This does not sufficiently alleviate my concerns on choice of random seeds.`
> > > >
> > > > Answer: To further address your concern, we tested with more random seeds in [40, 41, 42, 43, 44, 45], and provied the updated results in the tables below. Notably, we did not observe any consistent performance increase/decrease under certain random seeds, either in our method or in the baselines. From the tables we can see that our method still most often achieves the best task performance and fairness levels across the datasets.
> > > >
> > > > MNIST-USPS
> > > > | Methods     | Recall@1200       | ROCAUC         | Rec Diff       |
> > > > |-------------|----------------|----------------|----------------|
> > > > | FairOD      | 12.31±1.04     | 49.96±0.25     | 11.59±0.52     |
> > > > | DCFOD       | 12.65±0.31     | 50.15±0.21     | 8.98±0.71     |
> > > > | FairSVDD    | 15.57±1.24     | 58.31±1.14     | 13.71±2.20     |
> > > > | MCM         | 39.80±0.31     | 78.82±0.93     | 55.69±1.04     |
> > > > | NSNMF       | 39.10±0.76     | 65.24±0.67     | 62.82±4.17     |
> > > > | Recontrast  | 64.18±3.92     | 83.59±3.92     | 40.95±6.08    |
> > > > | FADIG       | 67.17±0.21 | 91.27±0.36 | 3.72±2.02 |
> > > >
> > > > COMPAS:
> > > > | Methods     | Recall@350       | ROCAUC         | Rec Diff       |
> > > > |-------------|----------------|----------------|----------------|
> > > > | FairOD      | 16.54±2.27     | 50.12±1.19     | 8.06±1.17      |
> > > > | DCFOD       | 16.01±2.31     | 49.63±1.24     | 9.80±1.36      |
> > > > | FairSVDD    | 15.32±1.84     | 52.64±4.37     | 11.56±3.21    |
> > > > | MCM         | 21.02±0.60     | 50.82±0.58     | 6.31±2.79      |
> > > > | NSNMF       | 22.91±0.26     | 57.96±0.43     | 36.14±1.14     |
> > > > | FADIG       | 34.41±0.28 | 61.68±0.51 | 5.89±3.93  |

---

> ### Comment · Reviewer_oKCG · 2024-11-25
>
> Thank you for your answers to my comments.
> I will reply point-by-point again.
>
> 1:
> Tabular data is by no means "easier" than computer vision data. Computer vision data, tabular data, time series data, etc. should in many cases just be considered separately.
>
> 2/3:
> Thank for you for elaboration. I would like to see this elaboration reflected in the final manuscript as well.
> - 1. Perhaps a better alternative to memAE would be the Normalized autoencoder (https://arxiv.org/abs/2105.05735). Please also include these experimental results in the paper.
> - 2. I do not see how using a vanilla autoencoder leads to less overfitting compared to different architectures.
> - 3. I am still unconvinced by the validity of this assumption. It would be better to either not assume this at all, or to substantiate the assumption further.
> - 4. I do not understand what the author means by this.
> - 5. I think it's extremely problematic that an assumption which is widely questioned is further propagated, even in recent works. Note that none of these works validate the assumption, so they do not add any weight to the assumption itself.
>
> 4:
> - If other authors in slightly different settings achieve substantially better results, than the manuscript should better reflect the change in settings which account for this difference.

---

> > ### Author Response · Authors · 2024-12-04
> > **Response to follow-up comments**
> >
> > Thank you for your follow-up comments. We are glad to provide the following answers to your further questions.
> >
> > `W1: Tabular data is by no means easier than computer vision data. Computer vision data, tabular data, time series data, etc. should in many cases just be considered separately.`
> >
> > Answer W1: We are not emphasizing that tabular data is easier than computer vision data. We were only trying to introduce some existing works of deep anomaly detection methods.  To avoid readers' misunderstanding, We have updated the statement in the introduction as "Up until now, a large number of deep anomaly detection methods have been introduced, demonstrating significantly better performance than shallow anomaly detection in addressing complicated detection problems in a variety of real-world applications such as computer vision tasks."
> >
> > `W2&3 Point 1: Perhaps a better alternative to memAE would be the Normalized autoencoder. `
> >
> > Answer P1: We tested the Normalized autoencoder on the four datasets and show the average results with std in the table below. We can see that the performance of Normalized AE is also very bad. Notably, on the MNIST-Invert dataset, it classifies all the samples from the protected group as the anomalies, which indicates a complete failure on such a group imbalanced dataset. We would like to include these results and our findings in the revision.
> > | Datasets| Recall| ROCAUC  | Rec Diff |
> > |----|----|-----|-----|
> > | MNIST-USPS | 50.38 (1.21)| 89.54 (0.20)| 45.81 (1.65)  |
> > | MNIST-Invert | 12.16 (3.16)| 54.66 (33.11)| 80.74 (26.14) |
> > | COMPAS| 26.37 (0.55)| 58.73 (0.89)| 22.15 (6.36)  |
> > | CelebA | 6.39 (3.40) | 34.92 (23.01)| 9.79 (3.46)   |
> >
> > `Point 2: I do not see how using a vanilla autoencoder leads to less overfitting compared to different architectures.`
> >
> > Answer P2: We meant that we tested on MemAE and found that a vanilla autoencoder leads to less overfitting. We provide the result of MemAE in the table below, and we can see that it has worse performance across the datasets.
> >
> > | Datasets | Recall | ROCAUC| Rec Diff|
> > |----|----|----|----|
> > | MNIST-USPS | 31.12 (6.69)| 66.07 (6.69)| 31.19 (14.81) |
> > | MNIST-Invert | 27.97 (7.73)| 77.27 (5.08)| 42.06 (41.58) |
> > | COMPAS | 20.15 (2.13)| 55.59 (2.89)| 20.10 (5.88)  |
> > | CelebA| 8.93 (1.29) | 52.85 (0.41)| 14.25 (4.61)  |
> >
> > `Point 3: I am unconvinced by the validity of this assumption.`
> >
> > Answer P3: In anomaly detection works based on auto-encoder, it is a very common assumption that the model is only capable of fitting normal samples but cannot fit anomalies, so that an input with a large reconstruction error is classified as an anomaly. This assumption also appears in many of the works you have mentioned in the reference. In our fair anomaly detection setting, the data consist of four subgroups: protected normal samples, unprotected normal samples, protected anomalies, and unprotected anomalies. Given that the model is supposed to only fit the normal samples (protected normal and unprotected normal samples), it is natural for us to assume the model is capable of fitting two out of the four subgroups. Otherwise, if the model is only capable of fitting one subgroup, e.g., protected normal, it will show unfair anomaly detection performance w.r.t. the protected and unprotected groups. If the model is capable of fitting more than two groups (i.e., it can fit both normal samples and anomalies), it cannot distinguish anomalies out of the normal samples. This explains why we assume that the model is capable of fitting two out of the four subgroups. And that motivates our design of $\epsilon$. We will add the detailed explanation to the revision.
> >
> > `Point 4: I do not understand what the author means by this.`
> >
> > Answer P4: We mean that compared with the complex backbone in other anomaly detection methods, one of the reasons why we chose the vanilla autoencoder is its efficiency.
> >
> > `Point 5: I think it's extremely problematic that an assumption which is widely questioned is further propagated, even in recent works. `
> >
> > Answer P5: The assumption that "Anomalies tend to exhibit large reconstruction errors" is the base assumption in the anomaly detection methods based on autoencoder. The references you provided are also based on this assumption, but with different designs of improvement with regularization.
> >
> >
> > `W4: If other authors in slightly different settings achieve substantially better results, then the manuscript should better reflect the change in settings which account for this difference.`
> >
> > Answer W4: The work [5] is supervised anomaly detection while our method is unsupervised anomaly detection. These two settings are not slightly different since we have no label information at all, and this increases the difficulty of anomaly detection. We would like to add discussion of the change of setting in the revision. It is common that one dataset can be used in both supervised tasks and unsupervised tasks and shows different performances, such as MNIST-USPS and COMPAS.

---

### Official Review · Reviewer_Nxv8 · 2024-11-03

**Soundness:** 2
**Presentation:** 2
**Contribution:** 2
**Rating:** 6
**Confidence:** 4

**Summary:**

The paper proposes a technique to introduce weighted/cost-based loss function that pays attention to underrepresented groups in data such that the dominant groups do not have undue influence on the learned model.

**Strengths:**

1. The proposed technique shows that if it is known what the underrepresented groups are, then we can possibly use that information to improve anomaly detection through fairly straight-forward ways

  2. The ablation experiments help to understand the importance of individual components of the loss function


============

Update after author rebuttals:

Revising my scores after going over other reviewer's comments and being satisfied with most of author responses to my comments.

**Weaknesses:**

Overall, the paper needs to be written more clearly and unambiguously. Main comments are:


1. Section 2: The paper defines protected groups on the basis of a single feature -- this might be rather simplistic. Instead, in real data, there could be a combination of one or more features that defines the protected/unprotected classification. In reality, it might not be easy to identify the underrepresented groups automatically.


2. Why restrict to a single protected group? There might be multiple 'protected' groups that have very distinct characteristics.


3. Wouldn't it be better to have separate anomaly detectors for each (prot/unprot) group? Was that approach considered as a benchmark?


4. Figure 2: It is hard to differentiate between red and pink dots in the figure.


5. Line 194: "... learning can alleviate this issue by pushing examples uniformly distributed in ..." -- Hard to understand this sentence. Maybe there is some grammatical error?


6. Lines 195-196: "... to learn representations by distinguishing different views of one example from other examples as follows:" -- Not clear what this means exactly.


7. Line 208: "... to maximize the cosine similarity between the representations of the two groups, ..." -- This is very ambiguous. It needs to be very clearly stated what the goal is: for example, maximize the similarity among the instances in the protected group? And separately among instances in the unprotected group? It seems that minimizing L_{FAC} in Eqn 3 implies minimizing the intra-group similarity (second term on the right). This has not been clearly stated. If there is more than one protected group, how would that affect the loss function?


8. Line 238, Eqn 5: It is not clear what exact \epsilon values were used in the experiments even though some selection technique is mentioned in the Appendix.


9. Line 354-355: "... improved by minimizing the discrepancy between the two distributions and ..." -- this discrepancy is probably a property of the real-world which is not in our control. Therefore, trying to minimize it should not be an option.


10. Line 375: "... method which incorporates the prescribed criteria into ..." -- Which prescribed criteria? Need to be more clear and specific.


11. Table 3: While the proposed technique (FADIG) is shown to perform well, it has an unfair advantage because it has the extra information about the under-represented groups that is not available to other algorithms.

**Questions:**

Please refer to the weakness section.

---

> ### Author Response · Authors · 2024-11-24
> **Reponse to Reviewer Nxv8 (1/2)**
>
> Thank you very much for your detailed and constructive comments. We are glad to provide the following responses to your questions.
>
> `Weakness 1&2: The definition of protected groups is rather simplistic, and it might not be easy to identify the underrepresented groups automatically. Why restrict to one single protected group?`
>
> Answer 1\&2: Our method can be easily extended to the multi-valued multi-group setting and it is not necessary for us to identify one single protected group from different subgroups. To be more specific, In our loss design, $L_\text{REC}$ is an auto-reweighted reconstruction error, and the weight is in proportion with $L_0^{G} - L_G$, where $G$ denotes a certain group. In the multi-valued, multi-group case, we can extend the design of $L_\text{REC}$ by weighting each reconstruction error of group $G$ with $\frac{L_0^{G} - L_G} { \sum_g L_0^{g} - L_g} $. Then, in $L_\text{FAC}$, $L_\text{fair}$ encourages the representation similarity between different groups, and we can extend it to the similarity between different combinations of multi-valued, multiple attributes. $L_\text{unif}$ encourages the uniformity within each group, and we can add the uniformity term for each group in the multi-valued, multi-group case.
>
>
> `Weakness 3: Comparison with separate anomaly detectors for each (protected/unprotected) group?`
>
> Answer 3: Thanks for the question. There are several reasons why we do not include it as a baseline in our work.
> 1. It's inefficient to train two separate anomaly detection detectors.
> 2. We do not know the ratio of anomalies in protected and unprotected group. If we train two separate anomaly detector model, it's hard to determine the number of anomalies in each group.
> 3. We also try to train two separate anomaly detectors for two groups. During the test time, we set the k for top-k selection in proportion to the group ratio (i.e., set top-k as k * protected group ratio for protected group model, and set top-k for unprotected group model as k * unprotected group ratio). In the following table we present the results (mean and std) of training separate detectors for different groups on the four datasets. Compared with this baseline, our method performs much better in both task performance and fairness. One possible explanation for the failure of the baseline is that separate models may overfit on certain groups and thus cannot distinguish anomalies from normal samples.
>
> | Dataset       | Recall       | ROCAUC       | Rec Diff     |
> |---------------|--------------|--------------|--------------|
> | MNISTandUSPS  | 63.87 (0.37) | 89.30 (0.96) | 19.73 (0.87) |
> | MNISTInvert   | 72.86 (0.00) | 96.80 (0.02) | 41.98 (0.00) |
> | COMPAS        | 15.02 (2.03) | 44.48 (3.09) | 18.60 (6.97) |
> | CelebA        |  8.22 (0.50) | 44.27 (0.91) |  0.91 (0.54) |
>
>
> `Weakness 4: Confusion of pink and red dots in Figure 2.`
>
> Answer 4: Thank you for the suggestion. We have updated Figure 2 in the revision.
>
> `Weakness 5: Confusion in Line 194.`
>
> Answer 5: Sorry for the confusion. We mean that 'encouraging uniformity with contrastive learning can alleviate this issue by pushing examples to be uniformly distributed in the unit hypersphere, as illustrated in Figure 2b'.
>
> `Weakness 6: Confusion in Lines 195-196.`
>
> Answer 6: In contrastive learning, for each data point, different "views" of the same instance are created to form positive pairs, while this instance forms negative pairs with other instances. Contrastive loss maximizes the alignment of positive pairs and the separation of negative pairs.
>
> `Weakness 7: Confusion in Line 208.`
>
> Answer 7: We have revised the statement as 'maximize the cosine similarity between the representations of the protected and unprotected group'. We have also added more detailed explanations in the revision. If there is more than one protected group, the loss function will still encourage the similarity among different groups while encouraging the uniformity within each group. To be more specific, $L_\text{fair}$ encourages the representation similarity between different groups, and we can extend it to the similarity between different combinations of multi-valued, multiple attributes. $L_\text{unif}$ encourages the uniformity within each group, and we can add the uniformity term for each group in the multi-valued, multi-group case.

---

> > ### Author Response · Authors · 2024-11-24
> > **Reponse to Reviewer Nxv8 (2/2)**
> >
> > `Weakness 8: Unclear what $\epsilon$ values were used.`
> >
> > Answer 8: In Appendix E.1 (line 820 and lines 830-832 in the original version), we provide our design of $\epsilon$. The value of $\epsilon$ changes with the losses $\mathcal{L}_0^U, \mathcal{L}_0^P$ during the training procedure.
> >
> > `Weakness 9:  Trying to minimize the discrepancy between the two distributions should not be an option.`
> >
> > Answer 9: Sorry for the confusion. We are trying to minimize the discrepancy between hidden representations of the samples from two groups, not the original data distributions.
> >
> > `Weakness 10: Unclear what prescribed criteria.`
> >
> > Answer 10: We have revised the manuscript in Lines 399-400.
> >
> > `Weakness 11: Unfair comparison because FADIG has extra information about the under-represented groups.`
> >
> > Answer 11: We would like to clarify that the group information is available to all the baseline methods, and our method does not necessarily need to identify which group is underrepresented. Additionally, we only utilize group information during the training phase. In the inference phase, we do not leverage group information for anomaly detection, e.g., manually controlling the ratio of potential anomalies for two groups.

---

> > > ### Comment · Reviewer_Nxv8 · 2024-11-26
> > > **Response to rebuttal 2/2**
> > >
> > > Weakness 8: Unclear what $\epsilon$ values were used -- The required information should be more visibly mentioned in the experiments section else it is easy to miss.
> > >
> > > Weakness 11: Unfair comparison -- While the authors mention that the information is available to other algorithms as well, it is not clear if those algorithms were designed to use that information. For a fair comparison, the baseline algorithms should be able to use the information during either or both training and inference time.

---

> > > > ### Author Response · Authors · 2024-12-04
> > > > **Response to follow-up comments**
> > > >
> > > > Thank you very much for your recognition of our work. We would like to provide the following answers to your follow-up quesitons.
> > > >
> > > > `W8: Unclear what $\epsilon$ values were used -- The required information should be more visibly mentioned in the experiments section else it is easy to miss.`
> > > >
> > > > Answer W8: Thank you for your suggestion. Aside from the statement of our design for $\epsilon$ in the proposed method design section in lines 262-265, we have also added this to the experiments section in lines 418-422 for more clear presentation.
> > > >
> > > > `W11: Unfair comparison. While the authors mention that the information is available to other algorithms as well, it is not clear if those algorithms were designed to use that information. For a fair comparison, the baseline algorithms should be able to use the information during either or both training and inference time.`
> > > >
> > > > Answer W11: For all the fairness-aware anomaly detection baselines we compare with, they are designed to use sensitive information during the training time. For other anomaly detection baselines, the sensitive information is included in the input data. Thank you for your suggestion. We would like to add these discussions in the training details.

---

> > ### Comment · Reviewer_Nxv8 · 2024-11-26
> > **Response to rebuttal 1/2**
> >
> > Thanks for responding to my comments in detail.
> >
> > Weakness 3: Comparison with separate anomaly detectors -- The results are very promising. Thanks.

---

### Official Review · Reviewer_VHqM · 2024-11-04

**Soundness:** 3
**Presentation:** 4
**Contribution:** 3
**Rating:** 6
**Confidence:** 5

**Summary:**

The paper focuses on fairness in outlier detection in unsupervised learning specifically addressing imbalance data that naturally arises in presence of minority groups. The paper presents a method for addressing representation disparity due to imbalance by proposing a fairness-aware contrative learning criterion and a reconstruction based network module with weights to account for patterns from minority groups. The work emphasizes the use fairness-aware constrastive loss as the main driver for framework as it leads to small total variation distance between the groups. Empirical results show the effectiveness of the proposed method across multiple real-world datasets when compared to exciting fairness-aware methods.

**Strengths:**

1. Rebalancing autoencoder with learnable weight for reconstruction loss is a simple way to encourage learning patterns from minority groups. I like the analytical calculation of \epsilon.
2. An elegant extension of contrasive entropy for uniform representation to incorporate fairness criterion as contrastive entropy across majority and minority groups.
3. Paper is easy to read and follow.

**Weaknesses:**

1. In most applications we have multi-valued multiple protected attributes. It seems that it is non-trivial to extend the loss function to cater to such datasets. How does the method scale with multi valued multiple protected attribute setup?
2. The choice of \alpha hyperparameter is arbitrary. Why 4 works and not 8? How does one choose the value of this hyperparameter in real world scenario?
3. The paper completely ignores the discussion on hyperparameter settings for the competitors. To ensure fairness to competitors, the detailed hyperparameter settings should be included for each compared method to better illustrate the benefits of the proposed method.

**Questions:**

Are there simpler considerations for \epsilon that can be informed from the imbalance factor of groups? How does \epsilon work for multi-valued attributes and multiple protected attributes?

---

> ### Author Response · Authors · 2024-11-24
> **Response to Reviewer VHqM**
>
> Thank you very much for your recognition of our work and your constructive comments. We are glad to provide the following responses to your questions.
>
> `Weakness 1: How does the method scale with multi-valued, multiple protected attributes setup?`
>
> Answer 1: In our loss design, $L_\text{REC}$ is an auto-reweighted reconstruction error, and the weight is in proportion with $L_0^{G} - L_G$, where $G$ denotes a certain group. In the multi-valued, multiple protected attributes case, we can extend the design of $L_\text{REC}$ by weighting each reconstruction error of group $G$ with $\frac{L_0^{G} - L_G} { \sum_g L_0^{g} - L_g} $. Then, in $L_\text{FAC}$, $L_\text{fair}$ encourages the representation similarity between different groups, and we can extend it to the similarity between different combinations of multi-valued, multiple protected attributes. $L_\text{unif}$ encourages the uniformity within each group, and we can add the uniformity term for each group in the multi-valued, multiple group case.
>
>
> `Weakness 2: The choice of $\alpha$ hyperparameter is arbitrary.`
>
> Answer 2: In Appendix F.4 we can observe that our method is robust to the choice of $\alpha$, and we show the results of $\alpha=4$ in the experiments. In real-world scenarios, since our method is robust to the choice of $\alpha$, there should be no much concern on how to choose $\alpha$. You can simply set $\alpha=4$ for convenience.
>
> `Weakness 3: Lack of the discussion on hyperparameter settings for the competitors.`
>
> Answer 3:  Thanks for the suggestion. In Appendix F.1, we include the training details. For all the competitors, we use suggested hyperparameter settings in their original papers and the official code implementations.
>
> `Question 1: Are there simpler considerations for $\epsilon$ that can be informed from the imbalance factor of groups? How does $\epsilon$ work for multi-valued attributes and multiple protected attributes?`
>
> Answer Q1:  In Appendix F.3, we compare our method with the variant which sets $\epsilon$ as the imbalance factor of groups. From Table 12 we can observe that our designed learnable $\epsilon$ can effectively better enhance task performance and meanwhile promote fairness. For multi-valued multiple attributes, as discussed in Answer to Weakness 1, we can set $\epsilon=\frac{L_0^{G} - L_G} { \sum_g L_0^{g} - L_g} $ for each subgroup $G$.

---

> > ### Comment · Reviewer_VHqM · 2024-12-01
> >
> > Thank you for your response.
> >
> > Weakness 2: I understand that \alpha can be conveniently set to a value of 4. However, I would like to understand the rationale for choosing "4" in the first place. More specifically, how do you decide the range of values and then pick up 4. A short discussion around this should be included in the paper.
> >
> > Weakness 3: The suggested hyperparameter settings may have worked for the choice of datasets the earlier works have used. However, this study has quite different datasets, and we do not know whether competitors argue that their recommended settings are generalizable. Therefore, I still think a proper discussion on hyperparameter selection for all the methods should be discussed. If necessary, additional experiments can be included in the appendix.

---

> > > ### Author Response · Authors · 2024-12-04
> > > **Response to follow-up comments**
> > >
> > > Thank you very much for your follow-up comments. We are glad to provide the following answers to your further questions.
> > >
> > > `W2: How do you decide the range of values and then pick up 4. `
> > >
> > > Answer W2: From our empirical study, we find that the two losses are in a similar range, thus we tested $\alpha$ with different small integers to ensure that there is no dominance of one loss over the other.  We choose $\alpha=4$ in the first place because we observe that it most often leads to the relatively top 2 best recall difference across different datasets, although our model is not sensitive to the change of $\alpha$.
> > >
> > > `W3: We do not know whether competitors argue that their recommended settings are generalizable. Therefore, I think a proper discussion on hyperparameter selection for all the methods should be discussed. If necessary, additional experiments can be included in the appendix.`
> > >
> > > Answer W3: For FairOD, We follow the instructions of the model configurations as proposed by their authors using grid search. For DCFOD, the author studied the hyperparameters $\alpha$ and $\beta$ and we set as their suggested values. FairSVDD suggested to set $\lambda=1$ for all datasets. And other baselines are tested insensitive to hyperparameters. Thank you for your question and we would like to add these discussions on the hyperparamter settings to the appendix.

---

### Official Review · Reviewer_7C9H · 2024-11-05

**Soundness:** 3
**Presentation:** 2
**Contribution:** 2
**Rating:** 5
**Confidence:** 4

**Summary:**

This paper addresses the challenge of ensuring fairness in anomaly detection (AD), particularly in scenarios with imbalanced data between protected and unprotected groups.  To tackle this, the authors propose FADIG, a fairness-aware anomaly detection method. FADIG integrates a fairness-aware contrastive learning module and a rebalancing autoencoder module to address fairness and data imbalance issues. Theoretical analysis demonstrates that the proposed method ensures group fairness, and empirical studies confirm its effectiveness and efficiency across multiple real-world datasets.

**Strengths:**

1. The issue of fairness in anomaly detection is a highly important and meaningful problem.
2. The paper provides theoretical guarantees for the proposed method.
3. The method's effectiveness is empirically validated on real datasets.

**Weaknesses:**

1. The paper presents two challenges: C1, handling imbalanced data, and C2, mitigating representation disparity. There is a strong coupling between these two challenges, as imbalanced data leads to representation disparity (i.e., group imbalance results in higher errors for protected groups, causing misclassifications). From this perspective, addressing C1 effectively resolves C2, which makes me question the necessity of the fairness-aware contrastive learning module.
2. The paper lacks novelty. The conclusion that group imbalance leads to accuracy parity issues is well-known, and addressing accuracy parity by solving group imbalance has been widely used. From the results presented in this paper, I do not see any specific uniqueness in the context of anomaly detection.
3. The paper lacks a discussion of related work, specifically in two areas:
* The differences between this work and existing work on anomaly detection and imbalanced data.
* The differences between the fair contrastive learning methods used in this paper and other existing methods.
4. The writing is poor. For example,
* the statement "In this work, we focus on the group fairness notion which usually pursues the equity of certain metrics among the groups. For instance, Accuracy Parity (Zafar et al., 2017) requires the same task accuracy between groups and Equal Opportunity (Hardt et al., 2016) requires the same true positive rate instead." suggests that the authors consider both Accuracy Parity and Equal Opportunity as fairness metrics. However, in the experiments, only Accuracy Parity is used, which is confusing.
* Figure 2 seems confusing.

**Questions:**

see weaknesses.

---

> ### Author Response · Authors · 2024-11-24
> **Response to Reviewer 7C9H**
>
> Thank you very much for your detailed and constructive comments. We are glad to provide the following responses to your questions.
>
> `Weakness 1: The strong coupling between the challenges of data imbalance and representation disparity makes the necessity of the fairness-aware contrastive learning module questionable.`
>
> Answer 1: We would like to clarify that the challenge of imbalanced data is two-fold: (1) the imbalance between the normal examples and the anomalies; (2) the imbalance between the groups. The fairness-aware contrastive learning loss consists of two terms, $L_\text{fair}$  and  $L_\text{unif}$.  $L_\text{fair}$ focuses on the imbalance between the protected and unprotected groups while $L_\text{unif}$ deals with the imbalance between normal examples and the anomalies by encouraging uniformity. In the ablation study in Section 5.4, we compare our method with the one without $L_\text{fair}$ (FADIG-N) and the one without $L_\text{unif}$ (FADIG-D), and replacing our designed fairness-aware contrastive learning loss with traditional contrastive loss (FADIG-C), and we can observe that our proposed method always has the best performance in both task performance and fairness, compared with all the variants. This validates the necessity of the fairness-aware contrastive learning module.
>
>
> `Weakness 2: Question of novelty in addressing accuracy parity by solving group imbalance in the task of anomaly detection.`
>
> Answer 2: In the task of anomaly detection, aside from the group imbalance, we are also faced with the imbalance between the anomalies and the normal samples. The mixture of the two types of imbalance makes the task of fair anomaly detection much more challenging. From the empirical comparison in Sections 5.2 and 5.3, we can see that our method usually outperforms all the existing fairness-aware anomaly detection baselines, which emphasizes the significance of our work in this field. In addition, we would like to further emphasize our contribution, including the novel re-balancing autoencoder, fairness-aware contrastive learning loss and the theoretical support for our proposed method.
>
> `Weakness 3: Lack of a discussion of related work`
>
> Answer 3: Please see our updated related work in Appendix B.
>
> `Weakness 4: Suggestions in writing and figure 2.`
>
> Answer 4: We have clarified the use of the fairness metrics in the revision to avoid confusion, and updated Figure 2 and its caption.

---

### Meta-Review · Area_Chair_oktA · 2024-12-13

**Metareview:**

I have read all the materials of this paper including the manuscript, appendix, comments, and response. Based on collected information from all reviewers and my personal judgment, I can make the recommendation on this paper, reject. No objection from reviewers who participated in the internal discussion was raised against the reject recommendation.

**Research Question**

This paper considers the fair outlier detection problem.

**Challenge Analysis**

The authors argue that the learning objectives of most existing anomaly detection methods tend to solely concentrate on the dominating unprotected group. To make this statement more concrete, the authors provide an empirical example in the introduction part. The authors analyze that misclassification arises from models focusing on learning frequent patterns in the more abundant unprotected group (A). Unfortunately, this statement has no rationality. Then, the authors aim to tackle two challenges (B), handling imbalanced data and mitigating the representation parity. Again, I do not see a link between (A) and (B). Moreover, the targeted challenges are the common tasks in fair outlier detection, where all the existing methods also consider these two points.

Here I would like to share my opinion on challenge analysis. For a newly defined research question, the challenge analysis comes from the difficulty of the research question; for a well-defined one, the challenge analysis comes from the drawbacks of existing solutions. For this paper, the authors seem to mix up the above points. Moreover, the presentation is not logically smooth.

**Technique**

For the above challenges, the authors propose two modules, fairness-aware contrastive learning module and re-balancing autoencoder. Technically speaking, I do not feel the proposed method is very novel, but it can tackle the research question.

**Theoretical Analysis**

The analysis in this paper is disconnected with the proposed method. I do not think it is a plus.

**Experiment**

The experiments seems extensive. However, two reviewers point out the comparisons to other methods and their experimental setup is insufficiently insightful. The authors need to demonstrate where the gain comes from.

**Additional Comments On Reviewer Discussion:**

No objection from reviewers who participated in the internal discussion was raised against the reject recommendation.

---

### Decision · Program_Chairs · 2025-01-22

Reject